# Boosting Visual-Language Models by Exploiting Hard Pairs

## Abstract

Contrastive Language-Image Pre-training (CLIP) has become the standard for learning cross-modal representations between images and text. Efforts to improve its capabilities typically demand the collection of additional data and retraining with new loss functions. While effective, the added requirements limit their practical use due to the increased resource and time investments needed. In this work, we present HELIP, a cost-effective strategy tailored to enhance the performance of existing CLIP models without the need for training a model from scratch or collecting additional data. Our method allows for effortless integration with existing models' training pipelines, providing an instant boost by training them with selected challenging text-image pairs from their original training datasets. HELIP treats each text-image pair as a single point in the joint vision-language space, identifying those in close proximity as hard pairs. By incorporating the challenging data, pre-trained CLIP models are refined using both the traditional contrastive loss and the newly introduced hard negative margin loss, ensuring the challenging data is fully utilized. On comprehensive benchmarks, HELIP consistently boosts existing models to achieve leading performance. In particular, it improves the zero-shot classification accuracy on ImageNet for SLIP models pre-trained on CC3M, CC12M and YFCC15M datasets. The improvements are 3.05%, 4.47%, and 10.1% respectively, achieved within two epochs of training. In addition, across fine-grained classification datasets, HELIP improves the zero-shot performance of pre-trained CLIP and SLIP by an average of 8.4% and 18.6%, and their linear probe performance by an average of 9.5% and 3.0%. The code is publicly available at `https://anonymous.4open.science/r/HELIP-7F8E/`.

## 1 Introduction

Contrastive Language-Image Pretraining (CLIP) (Radford et al., 2021) is quickly becoming hard standard for foundation models (Awais et al., 2023) due to its effectiveness for a variety of vision-language tasks without task-specific finetuning (Li et al., 2021; Baldrati et al., 2022). However, web-crawled image-text pairs used for the CLIP model pretraining are often loosely connected, resulting in multiple plausible matches beyond the assigned ones (Wu et al., 2022). Several methods have been presented to improve CLIP models by investigating appropriate matches and utilizing widespread supervision among image-text pairs for model training (Li et al., 2022a; 2021; Mu et al., 2022; Radenovic et al., 2023).

Efforts to improve contrastive language-image pretraining models have primarily taken two directions: (1) the addition of objectives to improve the efficacy of supervision (Li et al., 2022a; Mu et al., 2022); and (2) the employment of intra- and inter-modality similarity to select and retrain models using data deemed challenging at the sample level (Li et al., 2021; Radenovic et al., 2023). However, those approaches inevitably require retraining, and those identified as challenging data struggle to bring benefits to model performance. This challenge is partly due to their reliance on finding challenging data within a single batch during training, where truly beneficial challenging data is rare. Additionally, CLIP models' contrastive loss is not optimally configured to exploit the nuances of difficult data. These limitations significantly restrict the practical application of these enhancements, especially considering the substantial investments already made in pretraining numerous CLIP models (Li et al., 2022a; Mu et al., 2022). This aspect underscores the need for efficient enhancement strategies that do not rely on additional data collection to improve existing pretrained models.

To improve the existing CLIP models, we introduce the HELIP framework, which involves further training the models with challenging data selected from their original training dataset. HELIP defines and identifies the challenging data at the pair level, distinguishing it from traditional methods that compare sample-level similarities between images and texts. Specifically, HELIP treats each text-image pair as a distinct entity within the joint vision-language space, and defines pairs in close proximity as hard pairs. Furthermore, HELIP introduces the **Hard Pair Mining (HPM)** strategy, a novel approach that moves beyond the traditional use of representation spaces learned by CLIP models. Note, the CLIP space is primarily designed for evaluating sample-level similarities—for instance, comparing an image and text (individually, not as a pair)—lacking in evaluating characteristics at the pair level. HPM transforms the task of discovering pairs in close proximity into a solvable proxy task, with the goal of selecting a pair set that optimally supports the target pair's text-image agreement. HELIP enhances CLIP models not just with the original text-image contrastive loss (Radford et al., 2021), which uniformly pushes all negative samples away from their positive counterpart but also incorporates the **Hard Negative Margin Loss (HNML)** into the loss function. As depicted in Figure 2, HNML imposes an additional geometric structure on the representation space, reflecting the pair-level similarity. Through this approach, HELIP effectively leverages the information within challenging data to boost model performance.

Empirical evidence shows that HELIP improves the performance of existing CLIP models, including pre-trained CLIP, SLIP, and DECLIP, across a variety of benchmarks, such as zero-shot classification, text-image retrieval, and fine-grained linear probing. For zero-shot classification on ImageNet, CIFAR-10, and CIFAR-100, HELIP consistently boosts the performance of all six pre-trained models. Particularly, using HELIP to boost SLIP models pre-trained on CC3M, CC12M, and YFCC15M results in ImageNet zero-shot accuracy gains of 3.05%, 4.47%, and 10.14%, respectively. Further, on seven fine-grained image classification datasets, those pre-trained models achieve better zero-shot and linear probe performance with HELIP. Specifically, the average zero-shot accuracy of CC3M pre-trained CLIP and SLIP are improved by 8.4% and 18.6%. The average linear probe accuracy of CC3M pre-trained CLIP and SLIP are improved by 9.5% and 3.0% respectively. Additionally, the performance gain is also valid in terms of zero-shot retrieval, with 1.1 of R@1 on Flickr30K, and 2.2 of R@1 on COCO for SLIP with HELIP. Our contributions could be summarized as:

- To our best knowledge, our method, HELIP stands out as the first method aimed at improving existing CLIP models in a cost-effective and easily integrable way.

- We introduce the hard pair mining strategy to select challenging data, accompanied by the development of hard negative margin loss. This combination ensures the effective identification and utilization of challenging data, improving the CLIP models.

- Empirical evaluations across zero-shot classification, image-text retrieval, and linear probe benchmarks, consistently show HELIP's ability to substantially boost the performance of existing CLIP models, underlining its effectiveness and practicality in real-world applications.

## 2 Related work

**Vision-Language pre-training.** Vision Language Pretraining (VLP) is a technique that leverages large-scale image-text datasets to learn a strong joint representation between the two modalities that can be transferred to various downstream vision-language tasks. VLP models can be generally divided into single-stream models and dual-stream models. In the single-stream architecture, text and visual features are concatenated at the input level, creating a unified representation that is subsequently processed by a single transformer block (Li et al., 2019; Chen et al., 2022; Zhang et al., 2020). Conversely, dual-stream models (Jia et al., 2021; Li et al., 2022b; Mu et al., 2022; Radford et al., 2021; Yao et al., 2022) typically consist of two separate encoders for image and text respectively and perform cross-modality interactions on the top, are becoming more and more popular because of its flexibility of transferring pre-trained knowledge to downstream tasks. CLIP (Radford et al., 2021), uses a simple contrastive objective to learn visual features from natural language supervision and achieves remarkable zero-shot recognition performance using 400M web-crawled image-text pairs. Recent works boot the performance of CLIP by applying self-supervision within visual modal (Mu et al., 2022), additional nearest neighbor supervision (Li et al., 2022b). These

methods are actually doing data augmentations to increase data efficiency and thus bring additional computational costs.

**Contrastive learning with hard negative samples.** Contrastive learning learns a representation of input data that maps semantically comparable examples close together and semantically dissimilar examples far apart (Chen et al., 2020a;b; Wang & Isola, 2020). Recent works include hard negative samples into the loss function and achieve better empirical performance (Cai et al., 2020; Huynh et al., 2022; Kalantidis et al., 2020; Li et al., 2021; Radenovic et al., 2023; Robinson et al., 2021; Shah et al., 2022). For Language-image contrastive learning, current approaches (Li et al., 2021; Radenovic et al., 2023) mine multimodal hard negative examples using intra/inter-modality similarity. Li et al. (2021) choose in-batch hard negative samples with image-text contrastive loss. Hard negative noise contrastive multimodal alignment loss by Radenovic et al. (2023) up-weights the loss term for in-batch hard samples. For previous intra/inter-modality hard sample mining methods, two text-image pairs are considered as hard samples, if the cosine similarity between visual/textual features is high (Li et al., 2021; Radenovic et al., 2023). However, due to the nature of loose assignment for web-crawled image-caption data, a high similarity indicated by intra/inter-modality doesn't indicate that the two pairs are difficult to tell apart. Contrary to prior works, we design a hard sample mining method to discover similar pairs defined in joint vision-language space and efficiently select samples challenging enough to improve learning.

## 3    Hard pairs for visual-language models

In this section, we first define the notations and revisit CLIP for zero-shot recognition in the preliminary section. Next, we introduce the Hard Pairs Mining strategy (HPM), and the associated Hard Negative Margin Loss (HNML), specifically designed to leverage hard pairs identified by HPM.

### 3.1    Preliminaries

We consider the task of contrastive image-text pretraining. Given an image-caption dataset $\mathcal{D} = \{z_i\}_{i=1}^N = \{(x_i^I, x_i^T)\}_{i=1}^N$, $(x_i^I, x_i^T) \in \mathcal{I} \times \mathcal{T}$, the $x_i^I$, $x_i^T$ denote the image and its corresponding caption, $\mathcal{I}$ and $\mathcal{T}$ indicates visual and textual space respectively, and $\mathcal{I} \times \mathcal{T}$ indicates the joint Vision-Language space. The goal is to learn a dual encoder model $\phi = \{\phi_{image}, \phi_{text}\}$, where $\phi_{image}$ represents the image encoder and $\phi_{text}$ denotes the text encoder. We use the shorthand $I_i = \phi_{image}(x_i^I)$ and $T_i = \phi_{text}(x_i^T)$ to denote the encoded representation of an image and its caption, respectively. The contrastive objective of CLIP is formulated as,

$$\ell_{\text{CLIP}} = -\frac{1}{|B|} \sum_{i \in B} \log \frac{\exp\left(sim(I_i, T_i)/\sigma\right)}{\sum_{j \in B} \exp\left(sim(I_i, T_j)/\sigma\right)}, \tag{1}$$

where $sim(\cdot, \cdot)$ is the cosine similarity function, $B$ is a batch of samples and $\sigma$ is a trainable parameter controlling the temperature. Intuitively, the above formulation explicitly aligns the representations of image and text from one pair.

### 3.2    HPM: hard pair mining

In this study, we define *hard pairs* as the pairs that are nearby to a specified target pair within the joint vision-language space, $\mathcal{I} \times \mathcal{T}$, which serves as the domain for pair data. Equation 2 depicts the problem of hard pair mining. Here, $z_i$ represents the target pair, $\mathcal{H}_i$ denotes a set of pairs chosen from the dataset $\mathcal{D}_i = \mathcal{D} \setminus z_i$, and the metric $\mathbf{S}(,)$ quantifies the similarity between the target pair and a set of pairs,

$$\mathcal{H}_i^\star = \underset{\mathcal{H}_i}{\arg\max}\, \mathbf{S}(z_i, \mathcal{H}_i). \tag{2}$$

However, a key challenge arises in defining the similarity metric for pairs, $\mathbf{S}$. Existing CLIP methods (Radford et al., 2021; Li et al., 2022b;a) preliminary focus on aligning an image with its caption (Radford et al., 2021; Li et al., 2022a) from a image-text pair. They rarely emphasize on bringing similar pairs closer while distancing the dissimilar ones, which makes current methods fall short in gauging similarity between two pairs. For

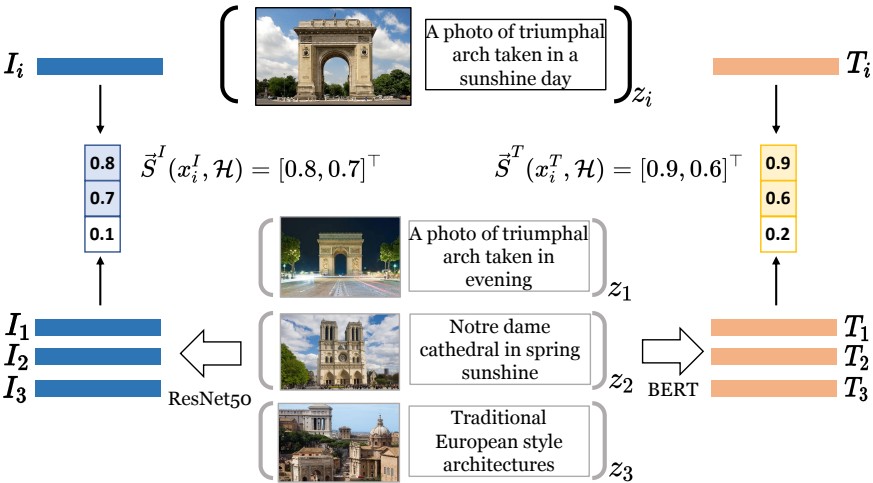

Figure 1: **Hard Pair Mining (HPM)**. Choose hard pairs by optimizing the support set to maximize the agreement prediction of the target pair.

instance, the cosine similarity between two pairs is ill-defined, within the context of current methods.

**Selecting hard pairs by maximizing pair agreement.** To identify nearby pairs, we introduce the idea of text-image pair agreement maximization. This can be viewed as a proxy task for selecting hard pairs. To illustrate the rationale for using text-image pair agreement as a proxy for selecting hard pairs, we return to the principle obtained from traditional machine learning methods: the prediction of a model on a test sample is substantially influenced by samples in the training dataset that are similar to the test one. For example, the K-Nearest Neighbors (KNN) algorithm classifies a new instance using the K-closest training examples. The linear regression model predicts the output of a test sample using the weighted sum of the training samples, with higher weights given to samples that are more similar to the test sample. Recent empirical and theoretical studies on model memorization and generalization (Chen et al., 2009; Zhang et al., 2021; Stephenson et al., 2021; Brown et al., 2021) also provide support for this. Intuitively, if a pair agreement prediction model trained on a set of pairs predicts a specific target pair as having a high probability of being a matching pair, the target pair is likely to be similar to the matching pairs on which the model was trained. The challenge of selecting hard pairs is transformed into an optimization task centered on the text-image pair agreement, which is formally represented as:

$$\arg\max_{\mathcal{H}_i} \mathbf{S}(z_i, \mathcal{H}_i) = \arg\max_{\mathcal{H}_i} P_{\mathcal{M}}(z_i | \mathcal{H}_i), \tag{3}$$

where $P_{\mathcal{M}}(z_i | \mathcal{H}_i)$ denotes the prediction of a pair agreement model, $\mathcal{M}$, for the pair $z_i$ based on a pair set $\mathcal{H}_i$. This set is a subset of $\mathcal{D}_i$. In this framework, the goal of selecting a hard pair is transformed into identifying a training set $\mathcal{H}_i$ such that the model $\mathcal{M}$ predicts the target pair as a matching pair.

Designing a suitable pair agreement prediction model for this proxy task is a nontrivial endeavor because the model needs to not only predict the pair matching probability but also allow the optimization of the training set, as indicated in Equation 3. Consequently, a conventional deep neural network design becomes unviable due to the impracticality of retraining across all possible sets $\mathcal{H}_i$ from $\mathcal{D}_i$. Taking inspiration from recent work (Norelli et al., 2022), we propose a data-centric design for the agreement prediction model $\mathcal{M}$. As illustrated in Figure 1, the model leverages two pretrained single-modal encoders, i.e., $f_{\text{image}}$ and $f_{\text{text}}$, to align representations of images and texts in a unified Vision-Language space. Specifically, the model encodes the target pair $z_i$ into $(I_i, T_i)$ using these single-modal encoders. For the visual modality, we determine a similarity vector between the target pair $z_i$ and the dataset $\mathcal{D}_i$. The similarity vector is defined as $\vec{S}^I(x_i^I, \mathcal{D}_i) = [\dots, sim(I_i, I_j), \dots]^\top \in \mathbb{R}^{N-1}$. Here $I_j = f_{\text{image}}(x_j^I)$ with $(x_j^I, x_j^T)$ being an element of $\mathcal{D}_i$, and function $sim(\cdot, \cdot)$ denotes the cosine similarity. To counteract noise, values in the vector $\vec{S}^I(x_i^I, \mathcal{D}_i)$ are set to zero if $sim(I_i, I_j) < \tau$. This cleaned-up vector is represented as $\widetilde{S}^I$. The procedure for the textual modality is analogous, producing a vector denoted as $\widetilde{S}^T$. Note, the representations in this shared space are intuitively interpretable: each dimension corresponding to the visual/textual similarity of the input to a

unique pair in the multimodal dataset. This interpretable characteristic enables us to directly optimize the supporting set to maximize the pair matching probability:

$$\mathcal{H}_i^{\star} = \arg\max_{|\mathcal{H}_i|=k} \widetilde{S}^I(x_i^I, \mathcal{H}_i)^{\top} \widetilde{S}^T(x_i^T, \mathcal{H}_i), \tag{4}$$

where the $\mathcal{H}_i^{\star}$ is the hard pair set and $k \in \mathbb{R}^+$ is the number of selected pairs which is much less than $|\mathcal{D}|$. The previous problem can be efficiently solved by greedily choosing dimensions that maximize the inner product. Due to the interpretable property, the selected dimensions correspond to the desired pairs.

**Mitigation of noisy data impact.** The prior method assumes the target pair $z_i$ to be a suitable matching pair. However, in inherently noisy datasets, such as web-crawled ones like LAION (Schuhmann et al., 2022), mismatched pairs might be present. The potential negative effects of hard pairs generated by these mismatched pairs necessitate the development of a strategy for identifying and eliminating them. We create a pair removal strategy based on the availability of hard pairs: A target pair $z_i$ is deemed as unsuitable and thus removed, if there is a non-empty subset of the mined hard pair set, $\mathcal{H}_i^{sub} \subseteq \mathcal{H}_i^{\star}$ with $|\mathcal{H}_i^{sub}| > 0$, such that $\widetilde{S}^I(x_i^I, \mathcal{H}_i^{sub})^{\top} \widetilde{S}^T(x_i^T, \mathcal{H}_i^{sub}) = 0$. Intuitively, this equation suggests that the number of entries positively supporting the target pair $z_i$ as a matching pair is fewer than $k$. To illustrate how this concept can aid in cleaning noisy data, consider the following example: Suppose the target pair consists of a "cat" image but a "dog" caption (clearly it is a mismatch). For it to be considered a correct match, numerous pairings with same erroneous pattern (i.e., "cat" images paired with "dog" captions) would need to exist in the dataset. By assuming a certain error types are fewer than $k$ throughout the dataset, if no subset of size $k$ within the dataset $\mathcal{D} \setminus z_i$ supports $z_i$ as a matching pair, this signals that the target pair is an outlier, likely due to a labeling error or mismatch. Such outliers can degrade dataset quality, so they are removed to ensure the reliability of hard data.

**Fast hard pair mining (FastHPM).** It is intuitive to infer that for a dataset collected from a single source, the number of intrinsic hard pairs, which are robust enough to enhance the learned representation, will proportionally increase with the size of the dataset originating from that source. To identify $k$ (much less than $|\mathcal{D}|$) qualified hard pairs, a portion of the dataset $\mathcal{D}$ is sufficient. As a result, we present the Fast Hard Pair Mining (FastHPM) approach, which was designed to avoid the time complexity associated with hard pair mining over the entire dataset. FastHPM's objective can be formalized as follows:

$$\mathcal{H}_i^{\star} \approx \arg\max_{|\mathcal{H}|=k} \widetilde{S}^I(x_i^I, \mathcal{H}_i)^{\top} \widetilde{S}^T(x_i^T, \mathcal{H}_i), \tag{5}$$

where $\mathcal{H}_i \subseteq \overline{\mathcal{D}}_i$ and $|\overline{\mathcal{D}}_i| = C$ is sampled uniformly from set $\mathcal{D}_i$. In this equation, it's noteworthy that the selection of value $C$ is solely based on the number of hard pairs $k$, instead of the size of $\mathcal{D}_i$. Consequently, this optimization reduces the time complexity of FastHPM to $\mathcal{O}(N)$. The detailed procedure of the hard pair mining algorithm is presented in Appendix A.

### 3.3 HNML: hard negative margin loss

The image-text contrastive loss $\ell_{CLIP}$, as illustrated in the preliminary section, aligns the true image-text pairs. But it poses no constraints on the overall geometry among data pairs (Goel et al., 2022). After involving hard data into the finetuning stage, equally maximizing the distance for normal negative pairs and hard negative pairs is an undesired way to utilize the information provided by hard negative pairs. The intuition follows directly from Figure 2. In a desired representation space, the similarity between the positive and the hard negative, $\mathbf{S}_1$, should be greater than the similarity between the positive and those normal negatives, $\mathbf{S}_2, \mathbf{S}_3$. Therefore, to impose the additional geometric structure, we introduce the Hard Negative Margin Loss (HNML):

$$\ell_{\text{margin}} = \frac{1}{|B|} \sum_{j \in B} \max\left(0, sim(I_i, T_j) - \min_{j' \in \mathcal{H}_i^p}\{sim(I_i, T_{j'})\}\right), \tag{6}$$

where $\mathcal{H}_i^p \subseteq \mathcal{H}_i^{\star}$ is the hard negative pairs for the target $z_i$ involved in one training batch. Note, the HNML is computationally efficient. No extra inner product computation is required. The geometric regularization

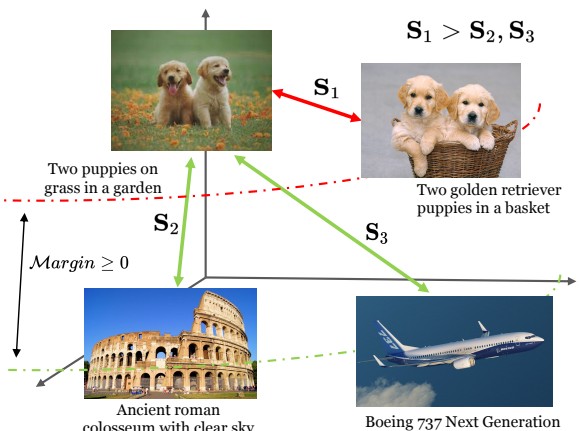

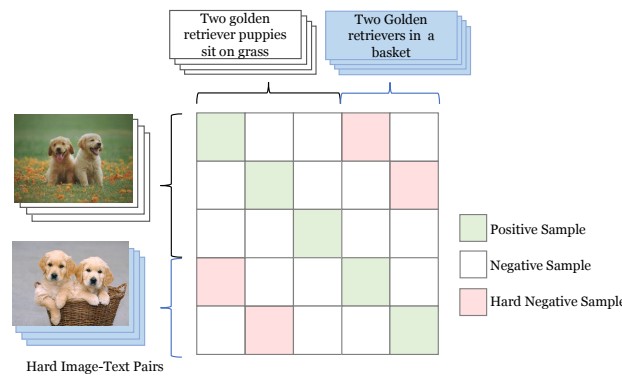

Figure 2: **Hard Negative Margin Loss (HNML).** Hard negative pairs are closer to the positive than the normal negative pairs.

Figure 3: **Further training CLIP with Hard Pairs.** For text-image pairs within a batch, we sample corresponding hard data from the preprocess hard pair set.

is applied over the inner product matrix computed in the original CLIP loss, Equation equation 1. Then, the well-trained model is finetuned with the following loss, where $\gamma$ is the hyperparameter balancing the two losses,

$$\ell_{\text{finetune}} = \ell_{\text{CLIP}} + \gamma \ell_{\text{margin}}. \tag{7}$$

To boost the performance of well-trained CLIP models without introducing extra data and extra parameters, we introduce the further training strategy which involves the preprocessed hard pairs into the batch composition during training. As shown in Figure 3, for text-image pairs within the batch $B$, we randomly sample a subset $B'$ as seeds. Then, for $z_i \in B'$, we randomly select $|\mathcal{H}_i^p| = p$ pairs from $\mathcal{H}_i^\star$. The actual training batch is $\overline{B} = B \bigcup_{i=0}^{|B'|} \mathcal{H}_i^p$. We summarize the training pipeline in appendix A.

## 4 Experiments

In the experiments, we conduct a comprehensive empirical investigation to evaluate the efficacy of HELIP in improving zero-shot classification, image-text retrieval, and linear probing performances for existing vision-language models, in Section 4.2. In Sections 4.3 and 4.4, we investigate HELIP's performance with scaled training data as well as its robustness over noisy datasets. We provide detailed empirical studies on Hard Positive Mining (HPM) and Hard Negative Mining with Margin Loss (HNML) in Sections 4.6 and 4.7.

### 4.1 Experimental setup

**Training datasets.** We used open-source datasets, including Conceptual Captions 3M (CC3M) (Sharma et al., 2018) and Conceptual Captions 12M (CC12M) (Changpinyo et al., 2021), and two 15M subsets of the YFCC100M dataset: v1, collected by Radford et al. (2021), and v2, collected by Li et al. (2022b). The combined datasets of CC3M, CC12M, and YFCC15M v1, which we denote it as Open29M following the term used in prior work (Li et al., 2022b), were not completely obtained due to expired urls. In addition, we independently sampled 7.5M and 8M subsets from the noisier data source, LAION-5B (Schuhmann et al., 2022), labeled as LAION7.5M and LAION8M. These datasets, while smaller than the 400 million pair dataset used in CLIP's original study (Radford et al., 2021), are well-suited for the data and computational resources we have. Furthermore, they have been widely used in benchmark evaluations for various studies on language-image pretraining, as noted in works by Goel et al. (2022); Li et al. (2022b) and Mu et al. (2022).

**Downstream datasets.** We primarily evaluate the effectiveness of HELIP using zero-shot image classification, linear probing, and zero-shot image-text retrieval. In addition to commonly used ImageNet (Deng

et al., 2009), CIFAR10, and CIFAR100 (Krizhevsky et al., 2009), we also verify the performance on 7 fine-grained classification datasets including Caltech101 (Fei-Fei et al., 2004), Food101 (Bossard et al., 2014), Sun397 (Xiao et al., 2010), Flowers102 (Nilsback & Zisserman, 2008), CUB (Wah et al., 2011), Stanford Cars (Krause et al., 2013) and FGVC Aircraft Maji et al. (2013). The zero-shot image-text retrieval task uses MS-COCO (Lin et al., 2014) and Flickr30K (Plummer et al., 2015).

**Implementation details.** Our experiments are conducted across three distinct architectures: ResNet-50, ViT-B/16, and ViT-B/32, tailored to various datasets and pretrained models. Specifically, for loading the pretrained CLIP model on CC3M and CC12M, the ResNet-50 is used as the image encoder. Besides, to align with existing checkpoints established by Mu et al. (2022), we use ViT-B/16 for SLIP model experiments on CC3M and CC12M, respectively. And, we use ViT-B/32 for pretraining on YFCC15M v1, v2, and Open29M datasets to ensure fair comparison with the results reported in Li et al. (2022b). Furthermore, for the SLIP and DECLIP models, we adapt the pretrained parameters from the publicly available resources* The input resolution of the image encoder is $224 \times 224$ and the maximum context length of the text encoder is 77. All of our experiments are conducted on 8 V100 GPUs with a batch size of 128 for ViT-B/16 models, and a batch size of 512 for ResNet-50 models and ViT-B/32 models. The dimension of the image and text embeddings is 1024 for ResNet-50 models and 512 for ViT-B/16 and ViT-B/32 models. We set $\tau = 0.5$, $\gamma = 1$, $k = 50$ and $p = 1$ for all the experiments by default. Automatic mixed-precision is used to save GPU memory. To keep the model from overfitting, we use early stopping if there is no performance gain on ImageNet zero-shot accuracy in 5 epochs. It is worth noting that using zero-shot classification performance on ImageNet as a criterion for early stopping is a commonly used practice for the training of CLIP (Radford et al., 2021; Mu et al., 2022). To reflect that our method is designed to work with few assumptions on encoder, we used encoders pretrained over a single-modal source rather than multimodally pretrained ones when preparing hard negative pairs. Specifically, we used an unsupervised pre-trained vision transformer, DINO VITs8 (Caron et al., 2021), and a Sentence Transformer (SentenceT) (Reimers & Gurevych, 2019) to encode text. For DINO VITs8, the embedding size is 384, while for SentenceT, it is 768.

## 4.2 Main results and discussion

**Zero-shot classification.** We compare the zero-shot performances of the CLIP, SLIP, DECLIP, and those models finetuned by HELIP on CC3M, CC12M, YFCC15M and Open29M. We denote the models finetuned by HELIP as CLIP-HELIP , SLIP-HELIP , and DECLIP-HELIP respectively. Table 1 demonstrates that models fine-tuned by HELIP consistently outperform their counterparts. Specifically, for models pretrained on the CC3M dataset, HELIP boosts the ImageNet zero-shot classification accuracy of the CLIP model from 19.04% to 19.86%. Additionally, on the SLIP model, a performance improvement of over 13% is observed, achieving an accuracy of 26.05%. We additionally include two baseline methods: CYCLIP (Goel et al., 2022) and CLOOB (Fürst et al., 2021) for reference. For CC12M pretraining, we used the SLIP checkpoints released by Mu et al. (2022). On ImageNet, SLIP-HELIP has a 4.47% higher zero-shot accuracy than its counterpart. Due to the lack of openly accessible parameters for DECLIP on the CC3M and CC12M datasets, our analysis focused on comparing DECLIP with DECLIP-HELIP over the YFCC15M v2 dataset. In this context, we present the performance of the SLIP and DECLIP models, as pretrained and released by Li et al. (2022b). The result was obtained by using their evaluation pipeline, denoted with ∗. Note, the templates are important for zero-shot tasks. Consequently, for a fair comparison, our analysis and conclusions primarily rely on results obtained from our evaluation pipeline, which is same as the approach used by OpenCLIP. Due to space constraints, we provide more information about the baselines in Appendix B. Notably, both SLIP and DECLIP showed improvements with HELIP, averaging increases of 15.49% and 6.74%, respectively. Further, to demonstrate HELIP's sustained efficacy across larger datasets, we assessed CLIP and CLIP-HELIP on Open29M. The original CLIP model, upon training with the Open29M dataset, achieves its best performance at the 18th epoch, achieving a zero-shot accuracy of 42.32% on ImageNet. The HELIP method instantly boosts the performance of the existing CLIP (checkpoint saved at the 18th epoch) from 42.32% to 46.33% with just one additional training epoch. However, extending the training with the original CLIP loss resulted in a marginal decline in accuracy to 42.25%.

---

*https://github.com/facebookresearch/SLIP, https://github.com/Sense-GVT/DeCLIP.

| | Method | ImageNet | CIFAR10 | CIFAR100 |
|---|---|---|---|---|
| CC3M | CYCLIP (Goel et al., 2022) | 22.08 | 51.45 | 23.15 |
| | CLOOB (Fürst et al., 2021) | 23.97 | - | - |
| | CLIP[†] (Radford et al., 2021) | 19.04 | 33.06 | 13.77 |
| | CLIP[†]-Helip | 19.86 | 34.05 | 14.13 |
| | SLIP (Mu et al., 2022) | 23.00 | 65.61 | 34.69 |
| | SLIP-Helip | **26.05** | **68.18** | **37.77** |
| CC12M | CLIP[†] (Radford et al., 2021) | 30.27 | 51.07 | 21.94 |
| | CLIP[†]-Helip | 32.05 | 52.27 | 24.51 |
| | SLIP (Mu et al., 2022) | 41.17 | 81.30 | 53.68 |
| | SLIP-Helip | **45.64** | **82.31** | **53.79** |
| YFCC15M | SLIP (Mu et al., 2022) | 25.29 (34.30*) | 60.19 | 26.80 |
| | SLIP-Helip | 35.43 | 75.49 | 47.84 |
| | DECLIP (Li et al., 2022b) | 36.05 (43.20*) | 78.12 | 50.60 |
| | DECLIP-Helip | **43.80** | **84.88** | **56.31** |
| 29M | CLIP[†] (Radford et al., 2021) | 42.32 | 71.98 | 42.73 |
| | CLIP[†] Cont. Train | 42.25 | 71.72 | 42.66 |
| | CLIP[†]-Helip | **46.33** | **77.97** | **48.33** |

Table 1: **Zero-shot classification performance on ImageNet, CIFAR10 and CIFAR100.** The †
indicates baselines pre-trained by us. For all other baselines, publicly available pre-trained parameters were
used. Specifically for SLIP and DECLIP on YFCC15M, we report results from two sources: our evaluation
using OpenCLIP's framework with pre-trained parameters released by Li et al. (2022b), and the performance
originally reported in Li et al. (2022b), marked with ∗.

**Zero-shot fine-grained classification.** By leveraging hard image-text pairs in contrastive learning, Helip
amplifies the discriminative capability of the CLIP model's visual embedding. This improvement proves
valuable in classification, particularly for fine-grained datasets. Our evaluation on 7 fine-grained classification
datasets (Table 2) reveals that SLIP-Helip boosts the zero-shot accuracy of CC3M and CC12M pretrained
SLIP on Caltech101 by 12.88% and 3.95% respectively. Both CLIP and SLIP models witness consistent
improvements with their Helip counterparts.

| Dataset | Method | Caltech101 | Food101 | Sun397 | Flowers102 | CUB | Stanford Cars | FGVC Aircraft | Average |
|---|---|---|---|---|---|---|---|---|---|
| CC3M | CLIP | 42.14 | 13.02 | 27.08 | 13.37 | 3.45 | 1.08 | 1.02 | 14.45 |
| | CLIP-Helip | 48.08 | 13.11 | 28.94 | 13.61 | 3.70 | 1.17 | 1.11 | 15.67 |
| | SLIP | 54.01 | 16.03 | 29.19 | 12.06 | 4.70 | **1.21** | **1.50** | 16.96 |
| | SLIP-Helip | **66.89** | **17.05** | **33.69** | **15.16** | **4.85** | 1.19 | 1.29 | **20.12** |
| CC12M | CLIP | 63.78 | 31.53 | 37.86 | 19.56 | 7.32 | 14.22 | 2.49 | 25.25 |
| | CLIP-Helip | 64.85 | 36.49 | 38.22 | 24.73 | 8.58 | 15.59 | 2.97 | 27.35 |
| | SLIP | 76.33 | 52.33 | 44.96 | **31.81** | 10.50 | 22.53 | 3.06 | 34.50 |
| | SLIP-Helip | **80.28** | **54.86** | **47.53** | 31.39 | **10.56** | **25.67** | **4.08** | **36.34** |

Table 2: **Zero-shot performance on fine-grained image classification.** On a variety of fine-grained
classification benchmarks, Helip consistent boosts the model performance compared to the original versions.

**Linear probing.** The linear probing task trains a randomly initialized linear classifier on the feature
extracted from the frozen image encoder on the downstream dataset. To accomplish this, we train the logistic
regression classifier using scikit-learn's L-BFGS implementation (Pedregosa et al., 2011), with maximum
1,000 iterations on those 7 datasets. For each dataset, we search for the best regularization strength factor
on the validation set over 45 logarithmically spaced steps within the range 1e-6 to 1e+5. Experimental
results in Table 3 demonstrate that both CLIP-Helip and SLIP-Helip have consistent improvements over
their counterparts on almost all 7 datasets. Note that on CC12M, SLIP-Helip performs marginally better
on 5 out of 7 datasets. It's probably because the self-supervision of SLIP (Mu et al., 2022) within the visual
modal can be beneficial for learning fine-grained visual embedding, while SLIP-Helip doesn't include image

self-supervision during the training. In addition, we did not match the training batch size as SLIP (Mu et al., 2022) because of resource limitations. A combination of HELIP and image self-supervision and larger training batch size may be a potential direction for achieving better linear probe performance.

| Dataset | Method | Caltech101 | Food101 | Sun397 | Flowers102 | CUB | Stanford Cars | FGVC Aircraft | Avg. |
|---------|--------|-----------|---------|--------|-----------|-----|--------------|---------------|------|
| CC3M | CYCLIP | 80.88 | 54.95 | - | 83.74 | - | 22.72 | 28.02 | - |
| | CLIP | 80.11 | 53.82 | 56.40 | 84.07 | 40.30 | 22.70 | 35.61 | 53.29 |
| | CLIP-HELIP | 82.49 | 59.79 | 59.56 | 87.84 | 46.19 | 30.01 | 42.48 | 58.34 |
| | SLIP | 87.96 | 72.50 | 66.96 | 91.91 | 49.77 | 39.25 | 45.87 | 64.89 |
| | SLIP-HELIP | **89.64** | **73.09** | **67.67** | **93.02** | **53.16** | **42.44** | **48.66** | **66.81** |
| CC12M | CLIP | 85.35 | 68.00 | 64.45 | 87.88 | 48.75 | 57.80 | 40.32 | 64.65 |
| | CLIP-HELIP | 85.87 | 68.89 | 64.95 | 88.36 | 49.41 | 58.55 | 40.17 | 65.17 |
| | SLIP | **92.89** | 83.63 | 74.34 | 94.87 | **60.99** | 73.43 | 52.23 | 76.05 |
| | SLIP-HELIP | 92.85 | **84.25** | **74.74** | **95.09** | 60.53 | **74.23** | **52.36** | **76.29** |

Table 3: **Linear probe performance on Fine-grained Image Classification.** On average, the linear probe performance of CLIP and SLIP pretrained on CC3M and CC12M are improved.

**Zero-shot retrieval.** We evaluate HELIP on zero-shot image-to-text retrieval tasks on MS-COCO (Lin et al., 2014) and Flickr30K (Plummer et al., 2015). As shown Table 4, both CLIP and SLIP, pre-trained on CC3M and CC12M , consistently improved by HELIP.

| Pretraining Dataset | Method | COCO | | Flickr30K | |
|---------------------|--------|------|------|-----------|------|
| | | R@1 ↑ | R@5 ↑ | R@1 ↑ | R@5 ↑ |
| CC3M | CLIP | 14.4 | 34.1 | 31.7 | 56.0 |
| | CLIP-HELIP | 17.8 | 39.8 | 35.4 | 61.0 |
| | SLIP | 22.3 | 45.6 | 39.6 | 68.6 |
| | SLIP-HELIP | **23.4** | **48.3** | **41.8** | **69.6** |
| CC12M | CLIP | 26.9 | 52.6 | 47.2 | 74.3 |
| | CLIP-HELIP | 27.8 | 54.3 | 48.2 | 75.4 |
| | SLIP | 39.0 | 66.0 | 65.4 | **90.1** |
| | SLIP-HELIP | **39.4** | **67.2** | **66.2** | 89.7 |

Table 4: **Zero-shot image-text retrieval results on MSCOCO and Flickr.** ↑ indicates higher is better. Combining with HELIP, CLIP and SLIP show better performance.

### 4.3 Performance of HELIP with Scaled Training Data

To investigate the impact of expanded training dataset sizes on the effectiveness of HELIP, we trained the CLIP model on the YFCC15M dataset. This training yielded a zero-shot classification accuracy of 25.46% on ImageNet. After applying HELIPand one epoch of training, its performance improved to 26.45%. To summarize the zero-shot performance on ImageNet of both the standard CLIP and its enhanced version, CLIP-HELIP, across different data scales, we have illustrated these results in Figure 4. The results show that HELIPconsistently enhances CLIP's performance. Most notably, the largest dataset, Open29M, witnessed a remarkable performance increase of 3.06% with HELIP. This result indicates that HELIPcan provide immediate performance enhancements for well-trained CLIP models on larger datasets, such as the private 400M dataset mentioned in Radford et al. (2021).

### 4.4 Performance of HELIP on noisy dataset

We expanded our investigation to assess the effectiveness of HELIPon subsets of LAION7.5M and 8M, which are randomly sampled from LAION (Schuhmann et al., 2022). The results are detailed in Table 5. The CLIP model, enhanced with HELIP consistently outperformed its original counterpart on both subsets across a majority of the evaluated datasets, including ImageNet, CIFAR10, CIFAR100, Caltech, and Food. On the

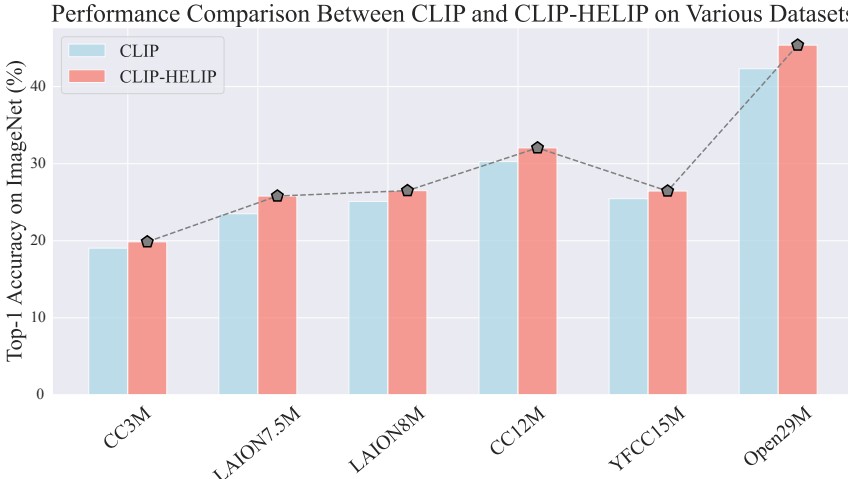

Figure 4: **Zero-shot performance on ImageNet for models pre-trained on different dataset sizes**.

7.5M subset, HELIP enhances performance across all datasets by an average of 3.6%. Although CLIP scores slightly higher on the Sun dataset, HELIPboosts its overall performance with an average improvement of 2.5% on the 8M subset. These results highlight the enhanced performance achieved through HELIP, demonstrating its robustness and effectiveness in improving existing models that have been pretrained on noisy data.

| | ImageNet | CIFAR10 | CIFAR100 | Caltech | Food | Sun | Avg. |
|---|---|---|---|---|---|---|---|
| CLIP-7.5M | 23.5 | 34.6 | 14.5 | 58.9 | 28.6 | 25.3 | 30.8 |
| CLIP-HELIP-7.5M | **25.8** | **39.9** | **16.7** | **61.9** | **34.1** | **28.2** | **34.4** |
| CLIP-8M | 25.1 | 31.1 | 12.9 | 60.9 | 29.5 | **27.5** | 31.2 |
| CLIP-HELIP-8M | **26.5** | **38.8** | **14.6** | **62.3** | **33.1** | 26.6 | **33.7** |

Table 5: **Zero-shot performance of CLIP on two LAION subsets.**

### 4.5 Comparison with other hard data selection method

We evaluate the efficacy of the proposed method in enhancing the discriminative capacity of learned representations by comparing its zero-shot classification performance with that of other hard data mining strategies. As described in the Section 2, a common way to define hard data is through intra-modality similarity. Hence, we introduce the hard data mining methods depending on (sample level) image similarity mining and text similarity mining and denote them as IM and TM respectively. For a given target pair, we compute the cosine similarity between its image/text representation and that of the remaining dataset. The image and text representations are encoded using a pretrained Resnet50 and BERT, respectively. As the preprocessing step, IM and TM methods mine hard negatives before continuous pretraining. Subsequently, we integrate the mined hard negative pairs into the training pipeline of CLIP and denote them as CLIP+IM and CLIP+TM and optimize the original contrastive loss to fine-tune the model. Additionally, we also include the hard negative contrastive loss, HN-NCE, proposed by Radenovic et al. (2023), as a baseline. HN-NCE upsamples the weight of hard-negatives identified by the current model. As shown in Table 6, when the CC3M pretrained CLIP model is combined with HELIP, the performance of our pair-level hard data mining method significantly outperforms other sample-level techniques. Besides,we observe that compared to the baseline CLIP performance, the introduction of TM and IM methods results in a decline in performance. To better understand the reasons behind this drop, we analyzed the outputs of the TM and IM methods in detail. In Figure5, we illustrate the data obtained through three distinct preprocessing methods: Hard Pair Mining (HPM), Image Similarity Mining (IM), and Text Similarity Mining (TM). The first row depicts the image-text pairs identified by HPM, while the second and third rows showcase the pairs mined by IM and TM, respectively. For TM (IM displays similar issues), the selected pairs often feature captions that are highly similar or identical, which is typical in data collected from the web. Even though identical pairs may not always be present, repetitions of the same images or text are common. According to

|  | Imagenet | CIFAR10 | CIFAR100 |
|---|---|---|---|
| CLIP | 19.04 | 33.06 | 13.77 |
| CLIP + TM | 16.70 | 28.71 | 9.67 |
| CLIP + IM | 16.93 | 29.22 | 10.42 |
| CLIP + HN-NCE | 19.47 | 29.88 | 11.83 |
| CLIP + HELIP | **19.86** | **34.05** | **14.13** |

Table 6: **Zero-shot performance of CLIP pre-trained on CC3M boosted by hard data mined by different methods.** HELIP shows superior performance, consistently outperforming local/global hard sample mining techniques by a substantial margin.

the CLIP contrastive loss (Equation 1), the model is forced to push nearly identical caption representations toward and away from two distinct image representations at the same time.This inherent contradiction in objectives contributes to a degradation in performance. To illustrate, consider a target pair $(T_{\text{target}}, I_{\text{target}})$ and a mined pair $(T_{\text{mined}}, I_{\text{mined}})$ using TM, where $T_{\text{target}} \approx T_{\text{mined}}$ but $I_{\text{target}} \not\approx I_{\text{mined}}$. In the contrastive loss framework, the model aims to minimize the distance between $(I_{\text{target}}, T_{\text{target}})$ and maximize the distance between $(I_{\text{target}}, T_{\text{mined}})$. However, the near-identity of $T_{\text{target}}$ and $T_{\text{mined}}$ leads to conflicting optimization targets and a potential decline in performance.

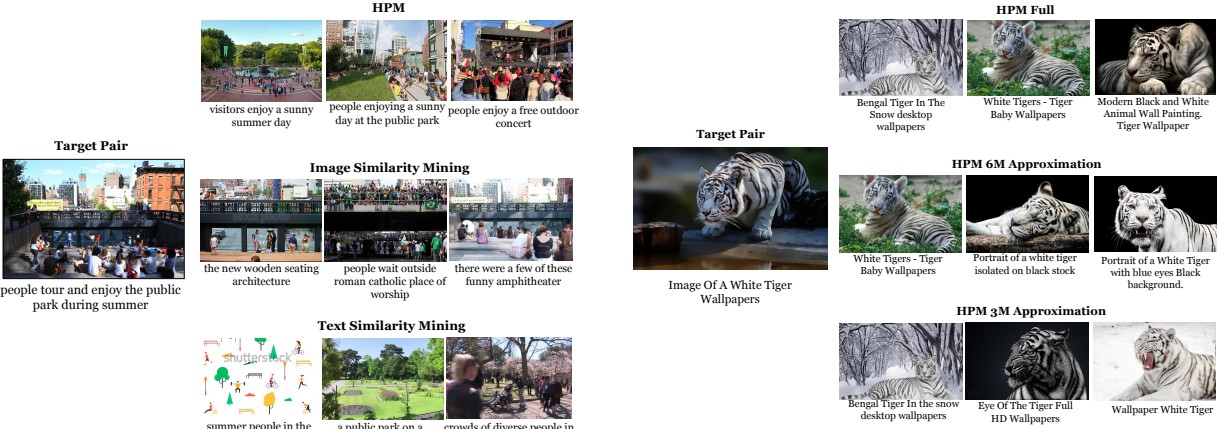

Figure 5: **Hard negative data selected by different methods.** Compared to data mined using the sample-level (image/text modal) similarity, hard pairs mined by HPM are more similar to the target.

Figure 6: **HPM and fastHPM.** We show the hard pairs mined by HPM and fastHPM. The quality of hard pairs mined by fastHPM is competitive with the pairs mined by HPM.

### 4.6 Impact of hard negative margin loss

We investigate the impact of using hard negative margin loss (HNML) on the performance of the SLIP model. In particular, our attention is directed towards an analysis of the SLIP model's performance, which has been previously pre-trained on the CC3M dataset, when it is both further trained with HPM+HNML and left without HNML. Our approach involves a comparative analysis of the model's zero-shot classification performance across multiple datasets including ImageNet, CIFAR 100, CIFAR 10, Caltech 101, Food 101, and Sun397. The results of our evaluation are comprehensively detailed in Table 7. These demonstrate that the SLIP model supplemented with HPM and HNML exhibits superior performance, with a performance boost of 4.51 and 3.27 compared to the SLIP and SLIP + HPM models respectively. Interestingly, the model achieved superior performance on the CIFAR 10 dataset without HNML. We postulate that this may be attributed to HNML's ability to enhance the discriminative power of the learnt representations by employing the class distance as a cost metric. In light of this, our findings suggest that for classification datasets consisting of a larger number of subclasses, employing HNML during the training phase can lead to an increase in classification performance.

|  | ImageNet | CIFAR10 | CIFAR100 | Caltech101 | Food101 | Sun397 | Avg. |
|---|---|---|---|---|---|---|---|
| SLIP | 23.00 | 65.61 | 34.69 | 54.01 | 16.03 | 29.20 | 37.09 |
| wo HNML | 24.94 | **69.44** | 36.35 | 64.07 | 16.51 | 30.91 | 40.37 |
| w HNML | **26.05** | 68.18 | **37.77** | **66.89** | **17.05** | **33.68** | **41.60** |

Table 7: **SLIP finetuned with and without hard negative margin loss.** When finetuned with hard pairs, the zero-shot performance of CC3M pretrained SLIP can be further enhanced usingx HMNL.

|  | ImageNet | CIFAR10 | CIFAR100 | Avg. |
|---|---|---|---|---|
| CLIP Encoders | 19.57 | 33.28 | 13.53 | 22.12 |
| VITs8 + SentenceT | 19.86 | 34.05 | 14.13 | 22.68 |
| VITb16 + SentenceT | 19.62 | 35.53 | 14.67 | 23.27 |
| VITs8 + T5 | 19.61 | 33.99 | 13.82 | 22.47 |

Table 8: **The zero-shot performances of HELIP with different encoders in HPM.** HPM's performance is insensitive to the selection of encoders.

## 4.7 Delving into hard pair mining

**Impact of different encoders in HPM.** We explored the effect of different pretrained encoders on HPM's performance by alternating image and text encoders. Initially, the unsupervised pretrained DINO VITs8 (Caron et al., 2021) was paired with the SentenceT (Reimers & Gurevych, 2019) transformer, trained on over a billion internet-based sentences. This combination was later swapped for the SWAG VITb16 (Singh et al., 2022) and the T5 (Raffel et al., 2020). Additionally, experiments using OpenAI's CLIP model (Radford et al., 2021) multimodal encoders were conducted. Interestingly, as Table 8 suggests, the encoder choice seemingly has negligible impact on HPM's performance, likely due to the proficiency of current pretrained models in modeling intra-modal similarities. Moreover, the ability to use single-modal pretrained models and still achieve competitive or superior performance implies that there's no assumption of having access to a high-quality CLIP model, such as OpenAI's CLIP-400M.

**Performance Comparison between HPM and FastHPM.** A comparison was made between the zero-shot performances of SLIP models, further trained with hard pairs obtained from both HPM and fastHPM. This comparison, conducted under three different settings, was summarized in Table 9. Additionally, we established subsets $\widetilde{\mathcal{D}}_i$ of sizes 3M and 6M, and accordingly denoted HELIP with these subset sizes as HELIP-3M and HELIP-6M. Table 9 shows that the zero-shot performances of HELIP-3M and HELIP-6M remain competitive with the global HPM hard pair mining approach. These findings suggest that fastHPM offers an efficient strategy for hard pair mining, without compromising performance. Additionally, they hint at fastHPM's potential to scale up hard pair mining in larger pre-training datasets, a promising direction for future exploration.

|  | Imagenet | CIFAR10 | CIFAR100 |
|---|---|---|---|
| SLIP | 41.17 | 81.30 | 53.68 |
| HELIP- 3M | 45.07 | **82.42** | 55.22 |
| HELIP- 6M | 44.98 | 81.64 | **56.62** |
| HELIP- Full | **45.64** | 82.31 | 53.79 |

Table 9: **Zero-shot performance for SLIP + HELIP on CC12M with hard samples mined with HPM and fastHPM.** Compared with hard samples mined with HPM, the fast versions are competitive with the full version.

**Visual insights into HPM and FastHPM.** We took the initiative to visualize the hard pairs as identified by the aforementioned three methods. Within Figure 6, the leftmost image-text pairing is earmarked as the target. The pairs in the primary row are those selected via HPM. The subsequent rows, specifically the second and third, present image-text pairings identified by the 6M fastHPM and the 3M fastHPM methods, respectively. Through a comparative visualization, it's evident that the hard pairs pinpointed by fastHPM

bear a significant resemblance to the target pair. For readers keen on delving deeper, we've provided an extended set of visualization outcomes in Appendix F.

**Computational time analysis.** Table 10 provides a comparison of the computational time required by HPM and fastHPM. The hard negative pairs preparation times listed were measured on 8 V100 GPUs, with the exception of the $*$ symbol, which was measured on a single V100 GPU. Given its efficiency and the performance similarities observed in Table 9, fastHPM emerges as a compelling alternative to the full HPM method.

|  | CC3M | CC12M | YFCC15M |
|---|---|---|---|
| HELIP- 3M | - | 2h18min | 3h27min |
| HELIP- 6M | - | 5h3min | 6h19min |
| HELIP- Full | 1h9min$^*$ | 9h11min | 17h41min |

Table 10: **Preparation time for hard pairs.** FastHPM speeds up the hard negative pairs mining process.

## 5 Conclusion

In this study, we delve into boosting pre-trained CLIP models' performance by more adeptly utilizing their original training dataset. This initiative arose from the recognition of the loosely connected nature of web-crawled image-text pairs, which resulted in suboptimal data utilization due to conventional CLIP loss. Our framework, HELIP, introduces a cost-effective and easily integrable solution for improving existing model performance without extensive retraining or additional datasets. It selects hard pair data from their original training datasets and refines the existing models in a few epochs to immediately boost their performance. Specifically, HELIP treats each text-image pair as a single point in the joint vision-language space, defining those that are close together as hard pairs. The Hard Pair Mining (HPM) strategy effectively identifies challenging hard pairs. The Hard Negative Margin Loss (HNML) was developed to improve existing models by utilizing that hard data. Empirical evaluations across various benchmarks, such as zero-shot classification, image-text retrieval, and linear probing, demonstrate the effectiveness and efficiency of HELIP. We leave the discussion of future work in the appendix.

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

# A   Appendix: Algorithm

We summarize the Hard Pair Mining (HPM), the fast Hard Pair Mining (fastHPM) and the training pipeline of HELIP in Algorithm 1, 2 and 3 respectively.

---

**Algorithm 1:** Hard Pair Mining (HPM)

---

**Input:** Hard pairs number per sample $k$

Pretrained unimodal vision model: $f_{\text{text}}$

Pretrained unimodal vision model: $f_{\text{image}}$

Dataset $\mathcal{D} = \{(x_1^I, x_1^T), (x_2^I, x_2^T), \cdots, (x_N^I, x_N^T)\}$

Threshold for visual and textual modality $\tau_I$ and $\tau_T$

**Output:** Hard samples $\mathcal{H} = [\mathcal{H}_1, \mathcal{H}_2, \cdots, \mathcal{H}_N]$

**for** $i \in [1, N]$ **do**

    $\mathbf{s} \leftarrow [0, 0, \cdots, 0]^\top \in \mathbb{R}^N$

    $I_i \leftarrow f_{\text{image}}(x_i^I)$

    $T_i \leftarrow f_{\text{text}}(x_i^T)$

    **for** $j \in [1, N]$ **do**

        $I_j \leftarrow f_{\text{image}}(x_j^I)$

        $T_j \leftarrow f_{\text{text}}(x_j^T)$

        $\vec{S}_j^I \leftarrow \frac{I_i \cdot I_j}{\|I_i\|_2 \cdot \|I_j\|_2}$ **if** $\frac{I_i \cdot I_j}{\|I_i\|_2 \cdot \|I_j\|_2} > \tau_I$ **else** $0$

        $\vec{S}_j^T \leftarrow \frac{T_i \cdot T_j}{\|I_i\|_2 \cdot \|T_j\|_2}$ **if** $\frac{T_i \cdot T_j}{\|T_i\|_2 \cdot \|T_j\|_2} > \tau_T$ **else** $0$

        $\mathbf{s}_j \leftarrow \vec{S}_j^I \cdot \vec{S}_j^T$

    **end**

    $\mathcal{H}_i \leftarrow \arg\max(\mathbf{s}, k)$

    **if** $\exists j \in \mathcal{H}_i, \mathbf{s}_j = 0$ **then**

        $\mathcal{H}_i = \emptyset$     # Indicate noise sample

**end**

---

Note, in the inner for loop, shown in Algorithm 1, the image and caption representations will be repeatedly computed. To accelerate the hard pair mining and avoid unnecessary computational overhead, we compute and save the encoded image features and text features. Besides, the outer loop is parallelized in the implementation.

# B   Appendix: Discussion about baselines

In our experiments, we utilized CLIP, SLIP, and DECLIP as baseline models on CC3M, CC12M, YFCC15M, and Open29M datasets. To ensure our results are both compelling and reproducible, we primarily employed publicly available checkpoints as our baseline and rigorously tested the effectiveness of HELIP against these checkpoints. On CC3M, the checkpoint of SLIP model is released[†]. We enhanced its performance by applying HELIP which notably improved the zero-shot performance on ImageNet from 23.00 to 26.05. However, we noticed that the CLIP with ResNet50 on CC3M is missing. To address this, we undertook the pretraining ourselves. Our results were encouraging: the performance of our pretrained CLIP with ResNet50 achieved a score of 19.86, surpassing the 17.10 achieved by SLIP's CLIP with ViT-B/32 as reported in Mu et al. (2022). This outcome suggests the robustness of our implementation. Besides, consistent with several prior studies, we found that on smaller pretraining datasets, CLIP with ResNet50 outperforms CLIP with ViT-B. On the CC12M dataset, a similar situation arose: while the SLIP checkpoint was available, the CLIP model was absent, leading us to undertake its pretraining. On the YFCC15M (v1) collected by Radford et al. (2021), we trained the CLIP model. This resulted in a 25.46 score in the ImageNet zero-shot classification, closely aligning with the 26.10 outcome reported by Cui et al. (2022). Additionally, for the YFCC15M (v2) dataset

---

[†]https://github.com/facebookresearch/SLIP#results-and-pre-trained-models

---

**Algorithm 2:** fast Hard Pair Mining (fastHPM)

---

**Input:** Hard pairs number per sample $k$
Pretrained unimodal vision model: $f_{\text{text}}$
Pretrained unimodal vision model: $f_{\text{image}}$
Dataset $\mathcal{D} = \{(x_1^I, x_1^T), (x_2^I, x_2^T), \cdots, (x_N^I, x_N^T)\}$
Threshold for visual and textual modality $\tau_I$ and $\tau_T$
Candidate pool size $C$
**Output:** Hard samples $\mathcal{H} = [\mathcal{H}_1, \mathcal{H}_2, \cdots, \mathcal{H}_N]$
**for** $i \in [1, N]$ **do**
    Uniformly $C$ samples from Dataset $\mathcal{D}$, $\overline{\mathcal{D}}_i = \{(x_1^I, x_1^T), (x_2^I, x_2^T), \cdots, (x_C^I, x_C^T)\}$
    $\mathbf{s} \leftarrow [0, 0, \cdots, 0]^\top \in \mathbb{R}^N$
    $I_i \leftarrow f_{\text{image}}(x_i^I)$
    $T_i \leftarrow f_{\text{text}}(x_i^T)$
    **for** $j \in [1, C]$ **do**
        $I_j \leftarrow f_{\text{image}}(x_j^I)$
        $T_j \leftarrow f_{\text{text}}(x_j^T)$
        $\vec{S}_j^I \leftarrow \frac{I_i \cdot I_j}{\|I_i\|_2 \cdot \|I_j\|_2}$ **if** $\frac{I_i \cdot I_j}{\|I_i\|_2 \cdot \|I_j\|_2} > \tau_I$ **else** $0$
        $\vec{S}_j^T \leftarrow \frac{T_i \cdot T_j}{\|I_i\|_2 \cdot \|T_j\|_2}$ **if** $\frac{T_i \cdot T_j}{\|T_i\|_2 \cdot \|T_j\|_2} > \tau_T$ **else** $0$
        $\mathbf{s}_j \leftarrow \vec{S}_j^I \cdot \vec{S}_j^T$
    **end**
    $\mathcal{H}_i \leftarrow \arg\max(\mathbf{s}, k)$
    **if** $\exists j \in \mathcal{H}_i, \mathbf{s}_j = 0$ **then**
        $\mathcal{H}_i = \emptyset$    # Indicate noise sample
**end**

---

referenced in Li et al. (2022b), both SLIP and DECLIP pretrained parameters were made available by Li et al. (2022b), which we utilized directly as our baselines. On the larger dataset, Open29M, there was a lack of open-source pretrained checkpoints, prompting us to conduct the pretraining ourselves. Notably, the performance of our reimplementation (42.32) closely aligns with the results reported by Li et al. (2022b), indicating the effectiveness of our approach.

## C   Appendix: Analysis of the Impact of Subset Size on Hard Pair Selection in FastHPM

In the comparison of HPM and FastHPM detailed in Section 4.7, we explore the efficacy of using 3M and 6M subset sizes of the CC12M dataset in FastHPM for mining hard pairs. The result, Table 9, shows that with a reduced subset size as small as 3 million entries, mining hard pairs and further training with these pairs can boost CLIP to achieve competitive performance with full set for mining.

In this section, we delve deeper into the analysis of hard pairs mined by FastHPM across varying subset sizes. Based on the selection criteria defined by FastHPM (Equation 5), we denote the *selection criteria value* as $\widetilde{S}^I(x_i^I, \mathcal{H}_i^\star(j))^\top \widetilde{S}^T(x_i^T, \mathcal{H}_i^\star(j))$. Here, $\mathcal{H}_i^\star(\cdot)$ represents a pair within the set of hard pairs $\mathcal{H}_i^\star$, mined by FastHPM for a specified target pair $i$ under a given subset size. Additionally, the $j$ in $\mathcal{H}_i^\star(j)$ indicates the $j$-th hard pair within the set $\mathcal{H}_i^\star$. Note, a higher selection criteria value signifies a harder mined pair.

We present the average selection criteria values for top-k hard pairs in Figure 7. As depicted by the grey horizontal line, the average selection criteria values for the top-20 hard pairs selected by FastHPM-1.5M, the top-40 by FastHPM-3M, and the top-80 by FastHPM-6M all approximate 0.477. This figure indicates that a further reduction in the subset size might necessitate adjustments to the number of hard pairs sampled to preserve quality. For instance, in our experiments detailed in Table 9, we uniformly sampled hard pairs for training from the top 50 for HELIP-3M. As Figure 7 suggests, a sampling range of 10 for HELIP-1M might

---

**Algorithm 3:** Hard samplE for boosting contrastive Language-Image Pretrained models (HELIP)

---

**Input:** $\mathcal{D} = \{(x_1^I, x_1^T), (x_1^I, x_1^T), \cdots, (x_N^I, x_N^T)\}$
Hard Pair Mining algorithm, HPM()     # or the fastHPM()
Pretrained unimodal vision model: $f_{\text{text}}$
Pretrained unimodal vision model: $f_{\text{image}}$
Pretrained contrastive language-image model $\{\phi_{\text{image}}, \phi_{\text{text}}\}$
hyperparameters:
    Hard pairs number $k$
    Hard negative margin strength $\gamma$
    Sampled hard negatives number $p$
    Learning ratio $\eta$
    Batch size $b$
    Training iteration number $E$
    Visual and textual modality threshold $\tau_I$ and $\tau_T$
**Output:** CLIP model $\{\phi_{\text{image}}, \phi_{\text{text}}\}$

$\mathcal{H} \leftarrow \text{HPM}(\mathcal{D}, f_{\text{text}}, f_{\text{image}}, k, \tau_I, \tau_T)$
**for** $iter \in [1, E]$ **do**
    $B \leftarrow \{z_1, \ldots, z_b\} \overset{\text{i.i.d.}}{\sim} Uniform(\mathcal{D})$
    **for** $z_i \in B$ **do**
        $\mathcal{H}_i^p \leftarrow \{z_i, \ldots, z_p\} \overset{\text{i.i.d.}}{\sim} Uniform(\mathcal{H}_i)$
        $\overline{B} \leftarrow B \cup \mathcal{H}_i^p$
    **end**
    Compute loss $\ell_{\text{finetune}}$, Equation (6), with samples $\overline{B}$ $\phi_{\text{image}} \leftarrow \phi_{\text{image}} + \eta \cdot \partial_{\phi_{\text{image}}} \ell_{\text{finetune}}$
    $\phi_{\text{text}} \leftarrow \phi_{\text{text}} + \eta \cdot \partial_{\phi_{\text{text}}} \ell_{\text{finetune}}$
**end**

---

be effective. Particularly, considering that HELIPsignificantly boosted the pre-trained models with just an additional training epoch, as discussed in Section 4.2, selecting one hard pair for each target pair from a pool of 10 will be feasible.

## D   Appendix: Analysis of the Impact of $\tau$ on Hard Pair Selection

To examine the impact of the threshold parameter $\tau$ on the selection of hard pairs, we analyze the similarities in the rankings of hard pairs (using Kendall Rank Similarity) mined by HPM under various $\tau$ values. The hard pairs are ranked by using the selection criteria value mentioned in Appendix C. The results on the CC12M dataset are displayed in Figure 8. We observe that the selection of hard pairs is robust to changes in the $\tau$ value. This resilience is partly because we only mine the top 50 hard pairs, a subset unlikely to be significantly affected when $\tau \leq 0.5$.

## E   Appendix: Analysis of the Impact of Mitigating Noisy Data

As presented in Section 3.2, to enhance the overall quality and reliability of the training dataset, data pairs lacking substantial support from the entirety of the training data are considered unsuitable and removed.

This section further empirically analyzes the impact of our noise mitigation strategy by detailing the quantity and nature of pairs removed across various datasets. Specifically, our approach removes 4.67% of the pairs from CC3M, 3.64% from CC12M, and 7.41% from YFCC15M, before continuing with pretraining. Figure 9 visualizes the pairs filtered from CC12M. Notably, our strategy effectively removed pairs such as unavailable images (e.g., two blank or white images in the second row) and mismatched pairs. These results suggest that

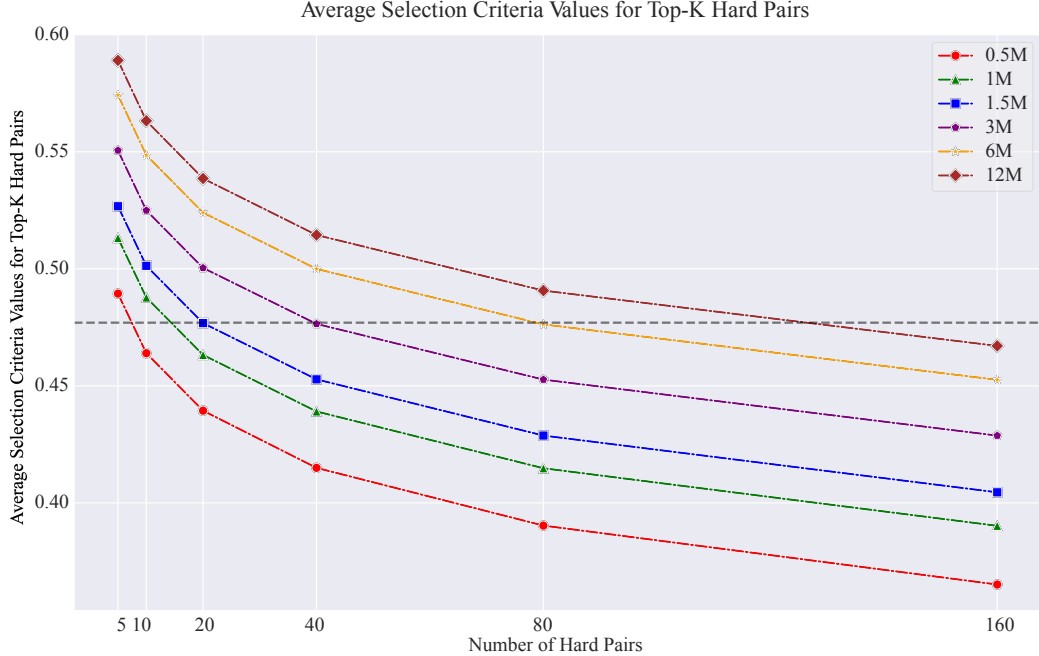

Figure 7: The average selection criteria values for hard pairs mined by FastHPM with different subset sizes.

our noise mitigation strategy can effectively clean the data using two single-modality models before training a CLIP model from scratch.

## F  Appendix: More visualization results

We offer further visualization results pertaining to the hard samples mined by various methods. As depicted in Figure 10, the hard samples sourced by HPM closely resemble the target sample (seen at the top left). Interestingly, for samples with fewer objectives, the image and text mining method can identify a reasonably challenging counterpart, as seen in the case of "the harbor in a small village". However, for intricate scenes, only the HPM is capable of yielding sufficiently challenging samples, like the scenario "people touring and enjoying the public park during summer". The dataset acquired from the web encompasses a myriad of such intricate cases. We posit that this is why training with hard samples unearthed by HPM yields more proficient outcomes.

Moreover, we present additional visualization results for hard samples mined via different techniques. Hard samples extracted by HPM exhibit a stronger resemblance to the target sample, as highlighted in Figure 10 (top left). We observed that the image and text mining methods can provide a relatively fitting hard counterpart for simpler samples, like "the harbor in a quaint settlement". However, for more intricate scenes, only the HPM method produces samples of adequate difficulty, such as "people touring and relishing the public park throughout summer". The web-based dataset includes a significant proportion of these complex cases. Consequently, we infer that training with hard samples mined by HPM results in enhanced performance.

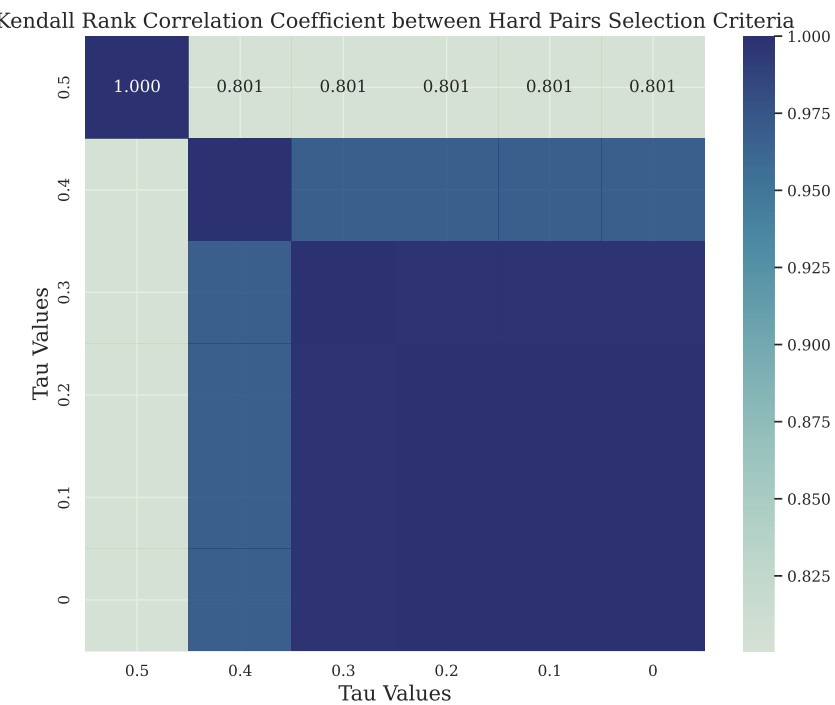

Figure 8: The Impact of $\tau$ on Hard Pair Selection.

# G   Appendix: Future work

Moving forward, several possibilities for future research emerge. First, we aim to explore composition-aware fine-tuning for VLMs, which could potentially enable more effective utilization of multimodal information. Moreover, we are intrigued by the prospect of combining parameter-efficient tuning (He et al., 2022) with HELIP potentially further enhancing performance. Another area of interest is scaling up the dataset size and examining the applicability of the scaling law to our method. We also intend to investigate how the integration of our boosting algorithm might alter the multimodal dataset curation algorithm (Gadre et al., 2023). Ultimately, we hope our work will serve as a catalyst for additional research in the fine-tuning of pre-trained, large-scale multimodal models.

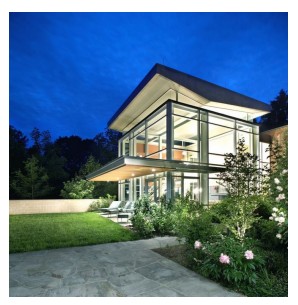

person have sent us images of person .

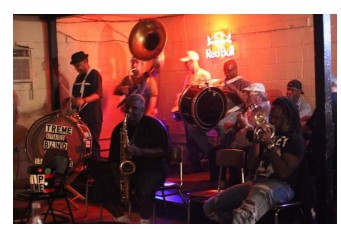

unfortunately , there are no appropriate pictures of us on the plane .

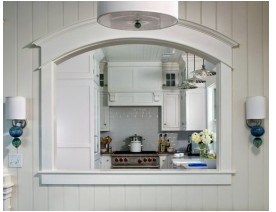

industry of turquoise , nice way to handle the pass through

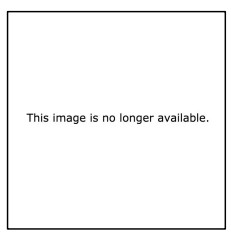

my nephew met my cat for the first time .

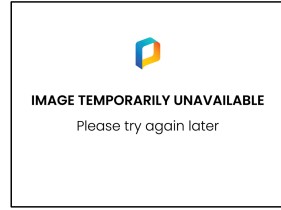

a male hiker looks over a view of granite mountains from the summit .

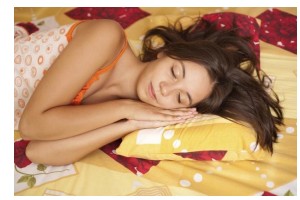

what should i do if i have bad cuts on the sides of my mouth ?

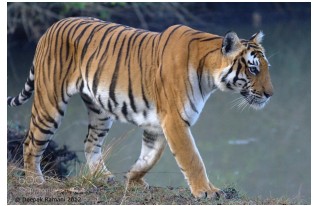

person - the lady by the lake

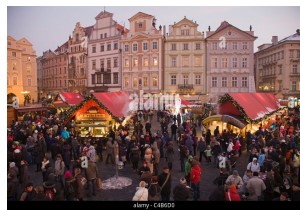

a street sign with the name of the place

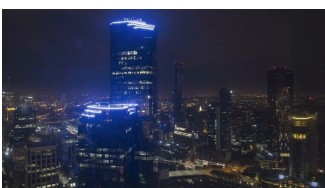

cloud , sun rays burst through the clouds during sunrise .

Figure 9: Visualization of the image-caption pairs filtered out from CC12M.

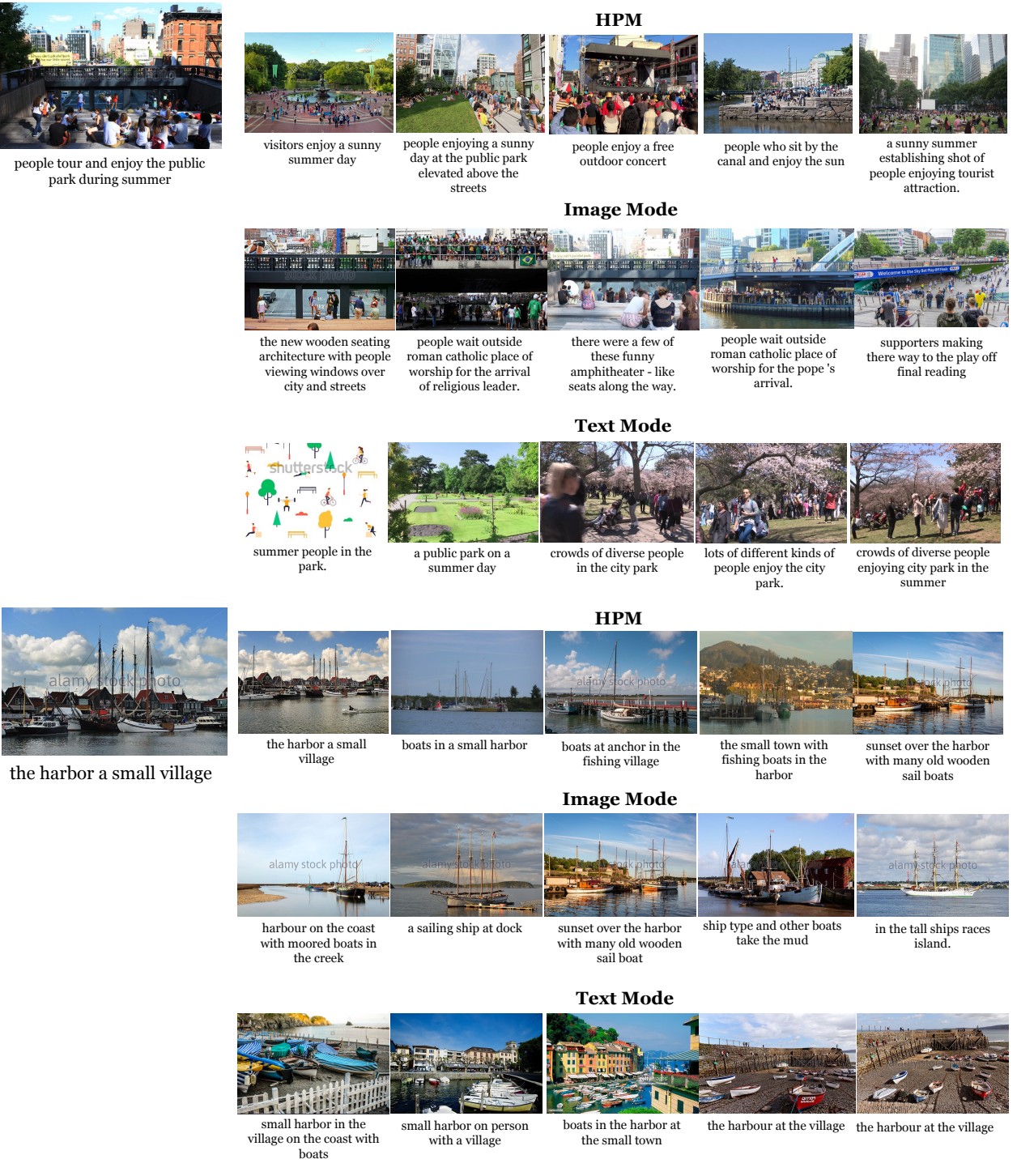

Figure 10: **Hard pairs selected by different methods.**

