# OpenReview forum: "Boosting Visual-Language Models by Exploiting Hard Pairs"
_TMLR — Rejected by TMLR_

### Review · Reviewer_da5i · 2024-04-04

**Summary Of Contributions:**

This work introduces Helip to enhance the performance of pre-trained CLIP models without necessitating additional data collection or model retraining from scratch. Helip identifies and utilizes challenging text-image pairs from the original training datasets, refining models with a combination of traditional contrastive loss and a newly proposed hard negative margin loss. The efficacy of Helip is demonstrated through improvements in zero-shot classification accuracy on ImageNet for SLIP models pre-trained on various datasets and notable gains in both zero-shot performance and linear probe performance across fine-grained classification datasets. These achievements highlight Helip's potential as a straightforward yet powerful tool for improving cross-modal representation learning with little additional resource.

**Audience:**

Yes

**Claims And Evidence:**

No

**Requested Changes:**

Please check weakness for details.

**Strengths And Weaknesses:**

Pros:
1. The proposed idea is intuitive and straightforward.
2. The paper is overall easy to follow and well presented.
3. Some of the results look good.

Cons:
1. The idea of hard negative mining is not really new to the community, as well as the point of mitigating need for finding global hard negative samples. Prior visual recognition/metric learning work has densely studied this problem [a,b,c,d]. It is important to comprehensively review the existing work and thus correctly position the work.

2. The other major concern is on the scalability of the method. Currently the method is only evaluated on rather small-scale models and it is more important that a model pretrained on very large scale dataset with a large size of parameters can still benefit from further tuning the model using the proposed method on a smaller scale data and computation.

3. The involvement of other pretrained models make it unclear whether there could be actually better way to distill their knowledge into the target model rather than simply using them to curate hard negative samples. It is also not clear whether it is necessary to use hard negatives during the training process in the proposed way, considering all the existing methods available, even with the ones that does not require hard negative mining [d].

[a] Facenet: A unified embedding for face recognition and clustering

[b] Training region-based object detectors with online hard example mining

[c] Smart mining for deep metric learning

[d] Deep Adversarial Metric Learning

---

> ### Author Response · Authors · 2024-04-10
>
> We thank the reviewer for their thoughtful feedback, highlighting that our work's idea is "intuitive and straightforward" and the "results look good". We address all comments below.
>
> **1. The Idea of Hard Negative Mining**
>
> ***(Related Work)*** We agree with you! A comprehensive review of existing work is important. We would like to thank you for the referred papers [a, b, c] that studied hard samples in face recognition, object detection, metric learning, etc.
> However, our work aims to study **the effect of hard pairs in CLIP**. Given that CLIP (or, more generally, contrastive learning) differs significantly from domains like face recognition, object detection, and metric learning, we sought to correctly position our work. Thus, we provided a detailed discussion on *"Contrastive Learning with Hard Negative Samples"* in Section 2 of the Related Work. This discussion includes studies on hard samples (a broader concept of hard pairs) in general contrastive learning and specifically in CLIP.
>
> ***(Global Hard Negative Samples)*** The "global hard pairs" is a key advantage of our proposed method, HELIP, setting it apart from the previous work mentioned. While methods that identify hard samples on the fly—thoroughly discussed in Section 2—struggle with several issues, HELIP excels by overcoming these challenges:
>
> 1. *Identifying Challenging Pairs*: HELIP efficiently explores hard pairs that are sufficiently challenging, which on-the-fly methods often fail to do.  These methods (as discussed in the related work section) attempt to identify a challenging sample in each iteration, which is hampered by time constraints, resulting in less effective discovery of truly difficult pairs.
>
> 2. *Computational Efficiency*: Unlike on-the-fly methods, which require repeated computations, HELIP operates more efficiently as  preprocessing.
>
> 3. *Integration with CLIP Framework*: HELIP is designed to integrate seamlessly with the current CLIPs' training pipelines, in contrast to on-the-fly methods, which may face integration difficulties.
>
>
> **2. The Scalability of the Method**
>
> Our method demonstrates scalability (no diminishment with larger dataset sizes) in experiments, as outlined in Section 4.3 Performance of HELIP with Scaled Training Data. The results show that HELIP consistently enhances CLIP’s performance across datasets of varying sizes, including CC3M, Laion 7.5M, Laion 8M, CC12M, YFCC15M, and Open29M. Notably, on the largest dataset, Open29M, we observed a remarkable performance increase of 3.06% with HELIP.
>
> Demonstrating scalability from small to large datasets is one of the most reasonable approaches at the current stage. When we considered using a larger LAION dataset, as suggested by the reviewer, to verify our method's performance, we discovered that the LAION dataset had been taken down due to the presence of CSAM, which is illegal (https://laion.ai/notes/laion-maintenance/, https://www.techpolicy.press/exposing-the-rotten-reality-of-ai-training-data/).

---

> ### Author Response · Authors · 2024-04-10
>
> **3. Distilling Involved Pretrained Models to Improve CLIP? Is It Necessary to Use Hard Negatives During the Training Process?**
>
> Distilling the involved pretrained models (visual and textual encoders) to improve CLIP is challenging, if not  outright infeasible. Firstly, it is important to note that the pretrained encoders used to mine hard pairs are single-modal. Employing these single-modal pretrained encoders as teachers, with a pretrained CLIP model as the student, complicates the design of the distillation method, necessitating further investigation.
>
> Moreover, as demonstrated in previous research [e] (refer to Figure 2 of Nguyen et al. [e]), the performance of single-modality pretrained models is generally inferior to that of CLIP. As a result, distilling the involved pretrained models to enhance CLIP is likely to result in a decrease in performance. In our method, we actually employ weaker (and smaller) models to encourage a stronger (and larger) model through hard pairs.
>
>
> Indeed, the use of hard negatives is essential in our study. Our research aims to explore the impact of hard pairs on CLIP. Considering alternative methods, such as the adversarial approach mentioned [d], to boost CLIP is considered beyond the scope of our current work.
>
>
> --
>
>
> [a] Schroff, F., Kalenichenko, D., & Philbin, J. (2015). Facenet: A unified embedding for face recognition and clustering. In Proceedings of the IEEE conference on computer vision and pattern recognition (pp. 815-823).
>
> [b] Shrivastava, A., Gupta, A., & Girshick, R. (2016). Training region-based object detectors with online hard example mining. In Proceedings of the IEEE conference on computer vision and pattern recognition (pp. 761-769).
>
> [c] Harwood, B., Kumar BG, V., Carneiro, G., Reid, I., & Drummond, T. (2017). Smart mining for deep metric learning. In Proceedings of the IEEE international conference on computer vision (pp. 2821-2829).
>
> [d] Duan, Y., Zheng, W., Lin, X., Lu, J., & Zhou, J. (2018). Deep adversarial metric learning. In Proceedings of the IEEE Conference on Computer Vision and Pattern Recognition (pp. 2780-2789).
>
> [e] Nguyen, T., Ilharco, G., Wortsman, M., Oh, S., & Schmidt, L. (2022). Quality not quantity: On the interaction between dataset design and robustness of clip. Advances in Neural Information Processing Systems, 35, 21455-21469.

---

> ### Comment · Action_Editor_nDkp · 2024-05-29
> **Make Replies Viewable to Reviewers**
>
> Dear Authors,
>
> It seems your replies above are only viewable to Editors and yourself. Reviewers can't see them.
>
> Could you please change the visibility to include all reviewers, or repost them?
>
> AE

---

> ### Author Response · Authors · 2024-05-30
>
> Dear AE and Reviewer da5i,
>
> We apologize for the oversight. We did not notice that the visibility settings were restricted. We have now changed the visibility of our post to include all reviewers.
>
> Thank you for bringing this to our attention.
>
>
> --
>
> Best regards,
>
> Authors of Submission 2323

---

> ### Comment · Action_Editor_nDkp · 2024-07-12
> **Official Recommendation**
>
> Dear reviewer,
>
> Could you please review the author response and kindly submit the official recommendation at the earliest convenience?
>
> The official recommendation is needed for us to move forward to subsequent steps.
>
> Thanks,
>
> AC

---

> > ### Author Response · Authors · 2024-07-12
> >
> > Dear Action Editor nDkp,
> >
> > Thank you for your support in advancing the reviewing progress of our manuscript.
> >
> >
> > Dear Reviewer da5i,
> >
> > Thank you for the time and effort you have invested in reviewing our manuscript. We have incorporated your valuable feedback into the revised version; please refer to the revision details. If you have any additional suggestions or comments, please share them with us so that we can improve our work even further!
> >
> >
> > Thanks,
> >
> > Submission 2323 Authors

---

### Review · Reviewer_mf4D · 2024-04-11

**Summary Of Contributions:**

The authors propose methods to finetune contrastively trained text-image embedding models using hard negative text-image pairs. Their contribution is two-fold:
* They introduce a method, HPM, to select hard negative pairs from the joint text-image datasets by relying on the representation of off-the-shelf image and text embedding networks.
* They introduce a hard negative margin loss (HNML) to leverage these hard negative pairs in enforcing that similar pairs should be closer in the embedding space than dissimilar pairs.

For a given text-image pair, HPM works by selecting for the hard negatives a subset of the text-image dataset for which a pair agreement model would predict maximum agreement for that pair. The pair agreement model is a dictionary method leveraging the encoding from the two frozen image and text embedders, and the selection of the maximizing subset is done greedily up to a fixed constant $k$. The authors additionally propose to do HPM on a randomly sampled subset of the dataset, rather than the full dataset.
HNML is a loss that ensures that the similarity of a pair is closer to the similarity of a hard negative than to the similarity of a random negative.

The authors show substantial improvement of their method in zero-shot and linear probe evaluation settings on popular datasets after fine-tuning of CLIP And SLIP with HPM or HPM + HNML, as well as a few ablation studies validating their method.

**Audience:**

Yes

**Claims And Evidence:**

Yes

**Requested Changes:**

1. Please address question 1. above.
2. Some experimental justiciation to address remark 2. above would strenghten the work.
3. Table 9 suggests that the dataset for sampling hard negatives can be significantly reduced without consistent degradation; it would be interesting to understand when this breaks, i.e. when is the dataset too small? (in the limit, this can become a random subset of negatives, similar to question 2).
4. I am not sure if the term "margin" is appropriate for the loss outlined in equation (6); indeed, for a "margin" to be present, I would expect a constant in the loss, e.g. $\mathrm{max}(0, 1 + \text{difference of similarities})$...
5. I would be interested to see ablations for the usefullness of $\tau$ (0 threshold - is this needed?)
6. I would also be interested to happen what the result look like without the "Mitigation of noisy data impact." startegy of removing "unsuitable" pairs.

Miscelaneous.

7. p.4: "such that the model M predicting the target pair as a matching pair" -> the end of the sentence is missing
8. Add CLIP performance in table 6.
9. p. 7: "we use early stopping if there is no performance gain in 5 epochs" -> how do the authors define this gain? Looking at the loss alone?
10. Remove repeated last sentence: :We leave the discussion of future work in the appendix. For the discussion of future work, we leave it in the appendix.:
11. Suggestion to add \text{} for text in equations, e.g. replace $\ell_{CLIP}$ by $\ell_{\text{CLIP}}$, $\ell_{margin}$ by $\ell_{\text{margin}}$, $f_{image}$ by $f_{\text{image}}$, etc... Similarly the notation argmax is inconsistent ($\mathrm{arg\ max}$ looks better than $Argmax$)

**Strengths And Weaknesses:**

Overall this looks like a solid work, very worthy of publication in TMLR.

+ The results are very convincing, with substantial improvements of the different metrics
+ The hard negative mining step can be done in a few hours on large subsets which seems very usable
+ The paper is well-written and the method is clear
+ The illustrations and ablations offer good insights in the effectiveness of the method

Some weaknesses / remarks
1. I am surprised by the results in Table 6: how can the performance of finetuning with CLIP + TM or CLIP + IM be so much lower than the baseline performance of CLIP alone (19.04)?
2. As a sanity check, I would like to see the performance of SLIP and CLIP finetuned with $\ell_{\text{CLIP}}$ under the same schedules without HNML or HPM, to validate that the performance is not due to extra training. Similarly, using a random subset of negatives rather than a mined subset could be a relevant baseline.

---

> ### Author Response · Authors · 2024-05-11
>
> We appreciate the reviewer's encouraging feedback. And thank you for dedicating time to carefully read our manuscript, identifying typos, and helping us improve the quality of our work!
> We are grateful for your comments on the "convincing" nature of our results and the "substantial improvements of the different metrics". We also value your recognition of the usability of the hard negative mining, which can be "done in a few hours on large subsets". This feedback confirms the practicality of our method. We address all the questions below.
>
> **Q1. Why does performance decrease when further finetuning CLIP with pairs selected vis Image similarity Mining (IM) or Text similarity Mining (TM) compared to the original CLIP alone?**
>
> The inappropriate supervision signals provided by pairs selected through Text Similarity Mining (TM) and Image Similarity Mining (IM) jeopardize performance.
>
> To illustrate, consider Figure 5 which focuses on TM (though IM displays similar issues). The selected pairs often contain captions that are highly similar or identical—a common occurrence in web-crawled data. Although identical pairs might not be present, the same image or text pieces often exist.
>
> This leads to a situation where, despite captions being nearly identical, their associated images differ. According to the CLIP contrastive loss (described in Equation 1 of our manuscript), the model is forced to simultaneously push the representations of nearly identical captions towards and away from two different image representations. This contradiction results in performance degradation. To formalize this:
>
> Consider a target pair $(T_{\text{target}}, I_{\text{target}})$ and a mined pair $(T_{\text{mined}}, I_{\text{mined}})$ using TM. Here, $T_{\text{target}} \approx T_{\text{mined}}$ but $I_{\text{target}} \not\approx I_{\text{mined}}$. In the contrastive loss, the model seeks to minimize the distance between $(I_{\text{target}}, T_{\text{target}})$ and maximize the distance between $(I_{\text{target}}, T_{\text{mined}})$. However, since $T_{\text{target}}$ and $T_{\text{mined}}$ are nearly identical, this results in conflicting optimization objectives.
>
>
> **Q2. How does the performance of further fine-tuning under the same schedule as original CLIP (without HNML or HPM), validate that improvements are not simply due to extra training?**
>
> We agree with the reviewer that assessing performance after additional fine-tuning using the original CLIP policy helps verify that improvements are not solely due to extended training. This approach also aids in demonstrating the effectiveness of our proposed method.
>
> Consequently, we further trained the CLIP model on the 29M dataset—the largest in our experiment and the one least likely to be prone to overfitting. We observed a decrease in performance from 42.32% to 42.25% on ImageNet, from 71.98% to 71.72% on CIFAR10, and from 42.76% to 42.66% on CIFAR100. We have updated Table 1 with the new results and provided a corresponding discussion highlighted in blue.
>
> Moreover, we believe that continuous pre-training with the best CLIP checkpoints will typically lead to a performance drop, especially on ImageNet zero-shot classification. As suggested in prior studies [1,2], the models achieving the best performance on ImageNet zero-shot classification are saved and reported. Consequently, for methods adhering to this standard protocol, additional training may result in a slight decline in performance.

---

> ### Author Response · Authors · 2024-05-11
>
> **Q3. Table 9 suggests that the dataset for sampling hard negatives can be significantly reduced without consistent degradation; it would be interesting to understand when this breaks, ie. when is the dataset too small? (In the limit, this can become a random subset of negatives).**
>
> Thank you for raising this question. We agree that exploring the limits of the subset size used by FastHPM could further enhance efficiency by determining the minimal size of the pair pool necessary.
> However, we would like to correct a misconception first. If the subset size is limited to one (a single pair in the subset), it would not represent "a random subset of negatives" but rather a deterministic hard pair repeated for each target pair in every batch. Consequently, when the pool is excessively small, it does not degrade into a scenario of random selection; instead, repeatedly using the same pair could potentially worsen performance.
>
> We acknowledge that developing a method to precisely predict the necessary subset size and the tradeoff curve between subset size and performance is non-trivial work. Solely relying on trial and error to adjust the subset size to see the CLIP performance is too ad hoc and lacks systematic depth, particularly for one or two datasets used in our work. Therefore, we would like to explore the design of systematic methods for predicting data volume requirements in future work.
> Here, we offer an empirical analysis of the hard pairs mined by FastHPM across various subset sizes. Detailed visualizations and discussions of this analysis are presented in Appendix.C of our revision (highlighted in blue text). We believe these discussions will be helpful in guiding and informing future research directions.
>
> **Q4. I am not sure if the term "margin" is appropriate for the loss outlined in equation (6); indeed, for a "margin" to be present, I would expect a constant in the loss**
>
> We agree with the reviewer's point about the expectation of a constant in "margin" losses. It's important to note that our definition of margin loss includes a constant 0, as shown in Equation (6). Our margin loss is intended to ensure that, within the representation space, the similarity between the positive and hard negative examples exceeds the similarity between the positive and typical negatives, thereby establishing a "margin" of differentiation.
>
>
> **Q5. The effect of \tau to the performance.**
>
> We investigated the effect of varying tau on the mined hard pair list. For ease of reference and to aid future readers, these results and detailed discussions are presented in Appendix D of our revision.
> Specifically, we employ Kendall's rank correlation coefficient to compare hard pair lists derived under different tau values. For the empirical study details and results, we would like to refer the reviewer to Appendix D.
>
> **Q6. I would also be interested to happen what the result look like without the "Mitigation of noisy data impact." startegy of removing "unsuitable" pairs.**
>
> Thank you for the suggestion. We conducted an empirical analysis of our strategy for mitigating the impact of noisy data. We believe this in-depth study enhances the quality of our work, so we have added a new section, Appendix E in our revision, to discuss these results.
>
> **Miscellaneous1: p.4: Original Text: "such that the model M predicting the target pair as a matching pair."**
>
> Thank you for your feedback. We have highlighted the correction in blue in the revision.
>
> **Miscellaneous2: "Add CLIP performance in table 6."**
>
> Certainly. We have included CLIP performance in Table 6 to enable easier comparisons for future readers and to more clearly demonstrate the effectiveness of HELIP's hard pair mining. Additionally, we have expanded the discussion of the updated Table 6 in Section 4.5.
>
>
> **Miscelaneous3: p. 7: "we use early stopping if there is no performance gain in 5 epochs" -> how do the authors define this gain? Looking at the loss alone?**
>
> Instead of loss, the performance gain is defined based on the model's performance on ImageNet. We acknowledge that using on-the-fly evaluation performance as a stopping criterion may seem unconventional from a traditional machine learning perspective. However, for CLIP training, monitoring model performance via ImageNet zero-shot classification evaluation is the most common approach, as suggested by [1, 2].
> We have recognized the necessity of clearly articulating this point. Consequently, we have elaborated it in the Implementation Details of Section 4.1, now updated and highlighted in blue for clarity.

---

> > ### Author Response · Authors · 2024-05-11
> >
> > **Miscelaneous4: Remove repeated last sentence: :We leave the discussion of future work in the appendix.**
> >
> > Thank you for your diligent review and for pointing out this redundancy. We have corrected it. Thank you for your attention to detail!
> >
> > **Miscelaneous5: Suggestion to add \text{} for text in equations, e.g. replace $\ell_{CLIP}$ by $\ell_{\text{CLIP}}$, $\ell_{margin}$ by $\ell_{\text{margin}}$, $f_{image}$ by $f_{\text{image}}$, etc... Similarly the notation argmax is inconsistent ($\mathrm{arg\ max}$ looks better than $Argmax$)**
> >
> > Thank you for your suggestion! We will make the following modifications:
> > - Replace $\ell_{CLIP}$ with $\ell_{\text{CLIP}}$ on pages 3 and 6.
> > - Replace $Argmax$ with $\mathrm{arg\ max}$ on pages 4, 18, and 19.
> > - Replace $\ell_{margin}$ with $\ell_{\text{margin}}$ on pages 5 and 6.
> > - Replace $f_{image}$ with $f_{\text{image}}$ and $f_{text}$ with $f_{\text{text}}$ on pages 4, 18, 19, and 20.
> >
> >
> > --
> >
> > Reference:
> >
> > [1] Radford, Alec, et al. "Learning transferable visual models from natural language supervision." International conference on machine learning. PMLR, 2021.
> >
> > [2] Mu, Norman, et al. "Slip: Self-supervision meets language-image pre-training." European conference on computer vision. Cham: Springer Nature Switzerland, 2022.

---

> ### Comment · Action_Editor_nDkp · 2024-07-12
> **Official Recommendation**
>
> Dear reviewer,
>
> Could you please review the author response and kindly submit the official recommendation at your earliest convenience?
>
> The official recommendation is needed for us to move forward to subsequent steps.
>
> Thanks,
>
> AC

---

> > ### Author Response · Authors · 2024-07-12
> >
> > Dear Action Editor nDkp,
> >
> > Thank you for your support in advancing the reviewing progress of our manuscript.
> >
> > Dear Reviewer mf4D,
> >
> > Thank you for taking the time to carefully read our manuscript, identify typos, and help us improve the quality of our work. We've incorporated your valuable feedback into the revised version. Please see the General Response or the revised PDF file for more information. If you have any further suggestions or comments, please share them with us so that we can improve our work even more!
> >
> > --
> >
> > Best regards,
> >
> > Submission 2323 Authors

---

### Review · Reviewer_EJU7 · 2024-04-29

**Summary Of Contributions:**

This paper mainly aims to boost the vision-language model, especially the CLIP.  Technically, the main idea lies in finetuning the pre-trained CLIP with hard text-image pairs to enhance the model's ability. Based on CLIP, a new model termed HELIP is proposed. HELIP contains a hard pair mining (HPM) strategy to find the difficult pairs and introduces the hard negative margin loss (HNML). Results are reported on zero-shot classification and also fine-grained classification datasets.

**Audience:**

Yes

**Broader Impact Concerns:**

/

**Claims And Evidence:**

Yes

**Requested Changes:**

1. discussion on related works could be more detailed. Take "vision-language pertaining" as an example, it mentions the single-stream and dual-stream models. However, no single paper nor discussion is given for the single-stream methods.
2. Fig.4 is in low-resolution.

**Strengths And Weaknesses:**

1. The paper is well-written and easy to follow.
2. The idea of re-using the hard pairs of training data for further boosting the performance of CLIP is simple, direct, and makes sense to me.
3. Extensive experiments are conducted and show the effectiveness of the proposed method.

---

> ### Author Response · Authors · 2024-05-11
>
> Thank you for your positive comments about our paper's clarity and structure. We are pleased that the concept of reusing hard pairs of training data to improve CLIP's performance arrived you as simple and logical. Furthermore, we appreciate your acknowledgement of our extensive experiments, which were critical in demonstrating our method's effectiveness. We address all the questions below.
>
>
> **Q1: discussion on related works could be more detailed.**
>
> Thank you for pointing this out. We have expanded our manuscript's discussion of related works on Vision-Language Pre-training (VLP). We've included discussions about single-stream models. The newly added sections （Sec.2 of revision） are highlighted in blue in the document for easy identification.
>
>
> **Q2: Fig.4 is in low-resolution.**
>
> Thank you for your feedback! We have redrawn Figure 4 to improve its resolution and overall quality. The updated figure should now be clearer and more visually appealing.

---

### Author Response · Authors · 2024-05-30
**General Response**

Dear Associate Editor and Reviewers,




We sincerely appreciate the time and effort invested by all reviewers in evaluating our paper! We apologize for the visibility issue with our reply to Reviewer da5i. Although we replied on April 10th, it was only visible to the Editors in Chief, Action Editors, and Authors. We have now fixed this issue.

In particular, we thank Reviewer mf4D for the comments about the performance of finetuning CLIP with TM/IM using the original CLIP training framework (continuous training), as well as various suggestions for writing. We have included our discussion results in the updated manuscript, which we believe improves our work!


For the convenience of reviewers and the action editor, below is a list of changes and their corresponding updates in the manuscript:

1. **More detailed discussion on related works** (*page 2*): We expanded our manuscript's discussion of related works on Vision-Language Pre-training (VLP) (Section 2).

2. **Figure 4 resolution** (*page 10*): We redrawn Figure 4 to improve its resolution and overall quality.

3. **Performance of finetuning with CLIP + TM or CLIP + IM** (*pages 10, 11*): We included the discussion about the performance in Section 4.5.

4. **Performance of continuous training of SLIP and CLIP finetuned with $\ell_{\text{CLIP}}$** (*pages 7, 8*): We updated Table 1 with the new results and provided a corresponding discussion highlighted in blue.

5. **Effect of the sub-dataset for sampling hard negatives** (*page 19*): Visualizations and discussions of this analysis are presented in Appendix C.

6. **Effect of \(\tau\) on performance** (*page 20*): We investigated the effect of varying \(\tau\) on the mined hard pair list and provide a discussion section in the appendix.

7. **Analysis of the impact of mitigating noisy data** (*pages 20, 21*): We conducted an empirical analysis of our strategy for mitigating the impact of noisy data.



--

Best regards,

Authors of Submission 2323

---

### Decision · Action_Editor_nDkp · 2024-09-09

**Recommendation:** Reject

**Comment:**

The AE has read the paper in its current form, the reviews, and all the discussions between reviewers and authors. In addition, the AE has also checked the previous TMLR submission and its associated reviews/discussions.

There is certainly good value and potential with the proposed method. The paper is easy to follow and the proposed method seems to have a sound design. A major concern, however, is regarding the scalability of the method and the true effectiveness of the gain. Reviewers, including da5i in this round, have pointed out whether the gain could diminish as the training scale goes up. While existing literature have indeed used smaller datasets such as CC3M, CC12M, and YFCC15M in the past, the contributions of this work are greatly weakened by the fact that recent modern CLIP models are mostly trained on LAION and DataComp datasets (the latter has not been found with CSAM problem). To some extent, this work seems a bit old-schooled by focusing on baselines from several years ago, which reduces the real value since the community has pretty much moved to much better CLIP models.

Of course, the AE is aware that the scale of experiment scale should not be the sole measure of the value for an academic submission. But as the authors suggest that their work is meant to refine CLIP models, there are many ways to show the effectiveness of the method in a more systematic and principled manner, as also suggested by the reviewers and AE. Unfortunately, there isn't a sufficient change from the last submission except for one additional CLIP Cont. Train baseline. The authors have not sufficiently addressed the concerns.

The AE would suggest the authors improve the experiments more comprehensively and resubmit to another venue. Several things could be considered as part of the revision:
1) Since the method is claimed to fine-tune CLIP models, the authors may include the experiments using recent OpenCLIP models trained on DataComp-1B, and continue the fine-tuning. A comprehensive pre-training is not needed if the resource is limited, since the method should immediately improve on top of the strong OpenCLIP baselines (original + continued training)
2) Include the continued training baselines for all the datasets beside Open29M (CC3M, CC12M and YFCC15M). Make sure that the continued training baselines strictly follow the original recipe with proper and sufficient number of additional iterations/epochs. Using the original baseline codebase with the method integrated would be the best.
3) Compare with other methods as stronger baselines. This is clearly not the first work to improve CLIP models with better losses, training recipes and hard negative mining. A search with Google already gives multiple related works, such as [a].

[a] Filtering, Distillation, and Hard Negatives for Vision-Language Pre-Training, CVPR23

**Audience:**

Audience in multi-modal training, representation learning, and transfer learning would find the work interesting to read.

**Claims And Evidence:**

In this paper, the authors present a method (HELIP) to improve the performance of Contrastive Language-Image Pre-training (CLIP) models by leveraging hard pairs in the training data. In particular, the method identifies challenging text-image pairs and introduces a new hard negative margin loss to refine pre-trained CLIP models, leading to improvements in zero-shot and fine-grained classification tasks. The paper demonstrates its effectiveness on various datasets, including improvements in zero-shot classification accuracy on ImageNet.

HELIP indeed leads to improved performance when added as an additional training stage to fine-tune pre-trained models using several relatively small-scale datasets. However, reviewer has raised concerns about scalability. For example, Reviewer da5i suggests that further examination on very large-scale datasets (larger than those tested) is necessary to truly confirm HELIP's scalability. Reviewer da5i also questions whether the source of HELIP’s gains is fully understood. While improvements are shown, there is a need for more careful ablation studies to disentangle the effects of the various components of HELIP. Hard Negative Selection: The authors claim that their method effectively selects challenging pairs, but reviewers suggested more clarity on how these pairs compare to randomly selected pairs or alternative methods without hard negative mining.